# Mathematical modelling and control of African animal trypanosomosis with interacting populations in West Africa—Could biting flies be important in main taining the disease endemicity?

**Paul Olalekan Odeniran**[1,2☯]*, **Akindele Akano Onifade**[3☯], **Ewan Thomas MacLeod**[2‡], **Isaiah Oluwafemi Ademola**[1‡], **Simon Alderton**[4‡], **Susan Christina Welburn**[2,5‡]

**1** Department of Veterinary Parasitology and Entomology, Faculty of Veterinary Medicine, University of Ibadan, Ibadan, Nigeria, **2** Infection Medicine, Biomedical Sciences, University of Edinburgh, Scotland, United Kingdom, **3** Department of Mathematics, Faculty of Science, University of Ibadan, Ibadan, Nigeria, **4** Centre for Health Informatics, Computing and Statistics (CHICAS), Lancaster Medical School, Lancaster University, Lancaster, United Kingdom, **5** Zhejiang University - University of Edinburgh Joint Institute, Zhejiang University, Haining, China

☯ These authors contributed equally to this work.
‡ These authors also contributed equally to this work.
* drpaulekode@gmail.com

## Abstract

African animal trypanosomosis (AAT) is transmitted cyclically by tsetse flies and mechanically by biting flies (tabanids and stomoxyines) in West Africa. AAT caused by *Trypanosoma congolense*, *T. vivax* and *T. brucei brucei* is a major threat to the cattle industry. A mathematical model involving three vertebrate hosts (cattle, small ruminants and wildlife) and three vector flies (Tsetse flies, tabanids and stomoxyines) was described to identify elimination strategies. The basic reproduction number ($R_0$) was obtained with respect to the growth rate of infected wildlife (reservoir hosts) present around the susceptible population using a next generation matrix technique. With the aid of suitable Lyapunov functions, stability analyses of disease-free and endemic equilibria were established. Simulation of the predictive model was presented by solving the system of ordinary differential equations to explore the behaviour of the model. An operational area in southwest Nigeria was simulated using generated pertinent data. The $R_0 < 1$ in the formulated model indicates the elimination of AAT. The comprehensive use of insecticide treated targets and insecticide treated cattle (ITT/ITC) affected the feeding tsetse and other biting flies resulting in $R_0 < 1$. The insecticide type, application timing and method, expertise and environmental conditions could affect the model stability. In areas with abundant biting flies and no tsetse flies, *T. vivax* showed $R_0 > 1$ when infected wildlife hosts were present. High tsetse populations revealed $R_0 < 1$ for *T. vivax* when ITT and ITC were administered, either individually or together. Elimination of the transmitting vectors of AAT could cost a total of US\$ 1,056,990 in southwest Nigeria. Hence, AAT in West Africa can only be controlled by strategically applying insecticides

**Data Availability Statement:** All relevant data are within the paper and its Supporting information files.

**Funding:** This study was supported by Commonwealth Scholarship Commission, United Kingdom. Paul O. Odeniran is a Commonwealth scholar, funded by the UK Government with reference number NGCN-2016-196. On our financial statement, I have submitted an application to PFA (PLOS Publication Fee Assistance), and the receipt of our application has been acknowledged.

**Competing interests:** The authors have declared that no competing interests exist.

targeting all transmitting vectors, appropriate use of trypanocides, and institutionalising an appropriate barrier between the domestic and sylvatic areas.

## Introduction

African animal trypanosomosis is a major constraint to sustainable livestock development in sub-Saharan Africa [1]. Tsetse flies (genus: *Glossina*) are the biological vectors of AAT caused by extracellular protozoan parasites of the genus *Trypanosoma*. The major causal organisms in livestock and wildlife are *T. congolense*, *T. vivax* and *T. brucei brucei*. There are other trypanosome species which are also of significant importance to the livestock herds such as *T. evansi* and *T. simiae*. There are complexities in the transmission dynamics of AAT. While *Trypanosoma brucei rhodesiense*, a common human pathogen has been incriminated to infect cattle in eastern Africa [2], the pathogen is absent in western Africa. More so, *Trypanosoma brucei gambiense* which affect humans in West Africa is rarely observed in livestock [3].

AAT is a well-known disease, causing devastating losses to the livestock industry which have been estimated to exceed US$ 1.3 billion annually [4–6]. Biting flies (families: Tabanidae and Muscidae) are effective mechanical vectors and are assumed to maintain AAT levels in various homesteads in sub-Saharan Africa [7]. In several countries there have been reports of AAT in cattle settlements free of tsetse flies, but where biting flies are abundant e.g. northern Nigeria, Cameroon and Chad [8–10]. The abundance of biting flies throughout the year is important in the epidemiology of the disease and parasite diversities among livestock hosts [11].

Human activities such as transhumance, settlement patterns, and vegetational changes have caused significant modifications in the vector habitat [6]. Recent natural occurrences such as persistent drought, landscape fragmentation, deforestation, environmental degradation, population pressure, and thinning out of wildlife, have been responsible for changes to the vector distribution map [6, 12–14]. These factors have significant importance on the epidemiology and control of AAT.

The elimination of vectors to control the disease has been focused on the biological vector, the tsetse fly, and most countries have engaged in prompt intervention strategies in the past such as aerial spraying of insecticides, sterile insect technique (SIT) and the most recent use of ethnoveterinary methods against the vector [15, 16]. Commonly used methods in West Africa are the use of insecticides (insecticide treated-targets (ITT) and insecticide treated-cattle (ITC)) and trypanocides to control the disease. However, the insecticide application strategy and methods have been reported to encourage resistance in biting flies [17]. The Pan-African Tsetse and Trypanosomiasis Eradication Campaign (PATTEC) in endemic areas is struggling to combat the disease in livestock in most countries since 2001, when it was initiated to eliminate tsetse in an area-wide approach [18].

Existing predictive models of AAT only consider the biological vectors *Glossina* spp., ignoring the possible importance of mechanical vectors [19–22]. The vector competence of *Trypanosoma* species in some tabanids and stomoxyines have been reported as a threat to the livestock industry [23–25], yet the elimination programmes often exclude these groups of flies (e.g. sterile insect techniques, aerial spraying of insecticides along tsetse pockets). Biting flies have been reported to display a wide range of activity patterns such as diurnal, nocturnal or crepuscular. Hence, various species are active at different times of the day [26], and this may have an indirect impact on resistance to insecticides.

The complexities of trypanosomosis transmission had limited the development of mathematical models, citing similarities with other vector-borne diseases like malaria [19]. Also, the few developed AAT models simulated from field conditions between susceptible cattle contracting trypanosomosis and tsetse infections had suffered as a result of knowledge gaps [20, 21]. This involves the inability to incorporate biting flies in field-based models because of its exclusion from the infectious group. However, the biting flies could be contaminated with *T. vivax* and mechanically transmit trypanosomes without being necessarily in the infectious group. These are important modelling factors to be considered in an all inclusive elimination approach. Several factors could affect the equilibrium conditions or the stability properties of AAT compartmental models considered (susceptible, exposed, infected, contaminated and recovered).

Therefore, this paper describes an agent-based model of *T. congolense*, *T. vivax* and *T. b. brucei* for the African animal trypanosomosis that incorporates three vertebrate hosts (cattle, small ruminants and wildlife), and three vector species (biological vector– *Glossina* spp. and two mechanical vectors-*Tabanus* spp. and *Stomoxys* spp.). We also evaluated the probability of the model in the preliminary report conducted in southwest Nigeria and explained the reality of its elimination. The developed model can easily be adapted in identifying crucial research priorities and providing several novel approaches in the control of AAT.

## Materials and methods

### Ethics

All protocols and procedures used in the field work were reviewed and approved by the University of Ibadan Animal Ethics Committtee with approval number (UI-ACUREC/App/12/2016/05).

### Model formulation

**Cattle population and modelling.** The model targets cattle as they play an important role in food security and can be seriously affected by AAT. The total cattle population size at time $t$ denoted by $N_k(t)$ is divided into susceptible cattle $S_k(t)$, exposed cattle $E_k(t)$, infected cattle $I_k(t)$ and recovered cattle $R_k(t)$. Hence, we have $N_k(t) = S_k(t) + E_k(t) + I_k(t) + R_k(t)$. For the cattle population: $\Lambda_k$ is the rate of new individuals entering the population, while $\mu_k$ and $\tau_k$ are the natural and disease induced death rates respectively. In the model, the terms $b\varphi_{k1} S_k I_t$, $b\varphi_{k2} S_k T_c$ and $b\varphi_{k3} S_k S_c$ denote the rates at which the cattle hosts get infected with AAT when they have contact with infected tsetse, $I_t(t)$, contaminated *Tabanids*, $T_c(t)$ and contaminated stomoxyines $S_c(t)$ with *T. vivax* (Table 1).

**Small ruminant population target.** Small ruminants (sheep and goats) develop clinical AAT during heavy challenge of trypanosomes. In endemic regions, small ruminants need to be treated to be in the recovered class. While those in the recovered class could return to the susceptible class if exposed to trypanosome infection in cases of persistent challenge, they rightly fit in the SIR model (susceptible, infected and recovered). Meanwhile, at the subclinical level of trypanosome infection, they are thought to maintain the infection in the domestic livestock cycle, serving as reservoirs for *Trypanosoma* species. This peculiar AAT eco-epidemiological situation in western Africa could largely be attributed to breed selection over the years (e.g. the West African dwarf goats and sheep) [16]. The total small ruminant population size at time $t$ denoted by $N_r(t)$ is divided into susceptible small ruminant, $S_r(t)$, infected small ruminant, $I_r(t)$ and recovered small ruminant, $R_r(t)$. Thus, we have $N_r(t) = S_r(t) + I_r(t) + R_r(t)$. For a small ruminant population, $\Lambda_r$ is the small ruminant input rate of new individuals entering the population. $\mu_r$ and $\phi$ are the natural and disease induced death rates respectively. The terms

**Table 1. Description of parameters.**

| Definition | Symbols | Value | Source |
|---|---|---|---|
| Cattle recruitment rate | $\Lambda_k$ | $\frac{1}{15 \times 365}$ | Estimated |
| Cattle loss of immunity | $\omega$ | $\frac{1}{12}$ | [19] |
| Natural death rate of cattle | $\mu_k$ | $\frac{1}{50 \times 365}$ | Estimated |
| Progression rate of exposed cattle | $\alpha_k$ | 0.516 | [3] |
| Proportion of effective treatment | $\rho$ | 0.12 | [19] |
| Recovery rate | $r_k$ | 0.1 | [19] |
| Death rate of tsetse fly | $\sigma_t$ | 0.212 | [19] |
| Progression rate of exposed tsetse fly | $\gamma_t$ | 0.028 | [31] |
| *Stomoxys* recruitment rate | $\Lambda_s$ | 0.075 | Estimated |
| *Tabanus* recruitment rate | $\Lambda_b$ | 0.0000548 | Estimated |
| Natural death rate of tsetse fly | $\mu_t$ | $\frac{1}{33}$ | [31] |
| Natural death rate of *Tabanus* | $\mu_b$ | $\frac{1}{33}$ | [3] |
| Flies contact rate | $b$ | $\frac{1}{14}$ | [3] |
| Transmission of infection to *Tabanus* | $\alpha_1$ | 0.014 | [31] |
| Transmission of infection to small ruminants | $\zeta_i, i = 1, 2, 3$ | 0.27 | Estimated |
| Death rate of *Tabanus* (insecticide) | $\delta_b$ | $\frac{1}{25}$ | [31] |
| Death rate of *Stomoxys* (insecticide) | $\delta_s$ | $\frac{1}{75}$ | [3] |
| Transmission of infection to *Stomoxys* | $\alpha_2$ | 0.031 | [31] |
| Transmission of infection to cattle | $\varphi_{k1}$ | 0.29 | [3] |
| Transmission of infection from *Tabanus* to cattle | $\varphi_{k2}$ | 0.33 | [31] |
| Small ruminants disease induced death rate | $\phi$ | 0.2 | Estimated |
| Transmission of infection to wildlife | $\tau_i, i = 1, 2, 3$ | 0.28 | Estimated |
| Small ruminant recovery rate | $\theta$ | 0.29 | Estimated |
| Small ruminant loss of immunity | $\theta_1$ | 0.78 | Estimated |
| Transmission of infection from *Stomoxys* to cattle | $\varphi_{k3}$ | 0.153 | [31] |

$b\zeta_{r1} S_r I_t$, $b\zeta_2 S_r T_c$, $b\zeta_3 S_r S_c$ denote the rates at which the small ruminant hosts get infected with AAT when they have contact with infected tsetse, $I_t(t)$, contaminated *Tabanids*, $T_c(t)$ and contaminated stomoxyines, $S_c(t)$ (Table 1).

**Wildlife population target.** Wildlife hosts are described in this model to maintain the disease in the domestic cycle, because they serve as reservoirs of *Trypanosoma* species. They rightly fit in the SI model (susceptible and infected). The total wildlife population size at time $t$ denoted by $N_w(t)$ is divided into susceptible wildlife, $S_w$, and infected wildlife, $I_w$. Thus, we have $N_w(t) = S_w(t) + I_w(t)$. For a wildlife population: $\Lambda_w$ is the wildlife input rate of new individuals entering the population and $\mu_w$ denotes the natural death rate. The terms $b\tau_1 S_w I_t$, $b\tau_2 S_w T_c$, $b\tau_3 S_w S_c$ denote the rates at which the wildlife hosts get infected with AAT when they have contact with infected tsetse, $I_t(t)$, contaminated tabanids, $T_c(t)$ and contaminated stomoxyines, $S_c(t)$ (Table 1).

**Transmitting vectors of bovine trypanosomosis.** Apart from feeding on cattle, it is assumed that tsetse flies can also be infected when feeding on infected wildlife and small ruminants with probability $\omega_1$ and $\omega_2$. *Tabanus* and stomoxyines cannot be infected, but contaminated with, *T. vivax* and then infect the cattle, small ruminants and wildlife hosts. For tsetse fly, tabanid and stomoxyine populations: $\Lambda_t$, $\Lambda_s$ and $\Lambda_b$ represent the input rate of new vectors entering the population. The total tsetse fly, tabanid and stomoxyine population sizes at time $t$ denoted by $N_t(t)$, $N_T$ and $N_S$ respectively, are divided into susceptible tsetse flies, $S_t$, exposed

tsetse flies $E_t$, infected tsetse flies, $I_t$, non-contaminated tabanids, $T_n$, contaminated tabanids, $T_c$ with *T. vivax*, non-contaminated stomoxyines, $S_n$ and contaminated stomoxyines, $S_c$ with *T. vivax*. Hence, we have $N_t = S_t + E_t + I_t$, $N_T = T_n + T_c$ and $N_S = S_n + S_c$. The terms $b\varphi_t\,\theta_1\,S_t(I_k + I_r + Iw)$, $\alpha_2\,\theta_2\,bS_n(I_k + I_r + Iw)$ and $\alpha_1\,\theta_3\,bT_n(I_k + I_r + Iw)$ denote the rates at which susceptible tsetse flies, non-contaminated stomoxyines and non-contaminated tabanids get infected or contaminated by infected cattle, small ruminants and wildlife species dominant in the area with *T. vivax*. Vector populations are affected by climate-dependent parameters [22], which is considered in the model. For instance, earlier models have established that pupae and teneral populations are low compared to adults in the very hot and wet rainy season [22]. However, at any given period, the ratio of pupae to mature tsetse was approximately 2:1, with mature tsetse to teneral at 15:1, growing as high as 25:1 when the general population is lower [22]. This in turn maintained the tsetse population over time. There are sex differences of tsetse to be considered in the model. An earlier report suggested a stable population growth in both the female and male tsetse flies [22]. Differences such as longevity, infectivity, mobility, and mortality changes in respect to age and responses to baits have been reported [20, 27, 28].

*Trypanosoma* **species of interest.** The pathogenic species of trypanosomes causing bovine trypanosomosis were considered in the model. The interactions of these species in both vertebrate and invertebrate hosts are important to the understanding of disease epidemiology. For the agent-based model, *T. congolense*, *T. vivax* and *T. b. brucei* were considered in the analysis. While the three parasites could be transmitted by tsetse flies, biting flies could only transmit *T. vivax*. All three parasites were considered with the assumption of its presence in the cattle, small ruminant and wildlife populations, provided the vector flies (tsetse flies and biting flies) are abundant. In the absence of tsetse flies, we incorporated biting flies, presence of wildlife species and examined the disease state (especially prevailing *Trypanosoma* species) in small ruminants and cattle herds. A schematic diagram of the model has been constructed (Fig 1). Based on the above assumptions, we have the following non-linear ordinary differential equations.

$$\frac{dS_k}{dt} = \Lambda_k - bS_k(\varphi_{k1}I_t + \varphi_{k2}T_c + \varphi_{k3}S_c) + \omega R_k - \mu_k S_k \tag{0.1}$$

$$\frac{dE_k}{dt} = bS_k(\varphi_{k1}I_t + \varphi_{k2}T_c + \varphi_{k3}S_c) - (\alpha_k + \mu_k)E_k \tag{0.2}$$

$$\frac{dI_k}{dt} = \alpha_k E_k - (\rho r_k + \tau_k + \mu_k)I_k \tag{0.3}$$

$$\frac{dR_k}{dt} = \rho r_k I_k - (\omega + \mu_k)R_k \tag{0.4}$$

$$\frac{dS_t}{dt} = \Lambda_t - b\varphi_t S_t(I_k + I_r + Iw) - (\mu_t + \sigma_t)S_t \tag{0.5}$$

$$\frac{dE_t}{dt} = b\varphi_t S_t(I_k + I_r + Iw) - (\gamma_t + \mu_t + \sigma_t)E_t \tag{0.6}$$

$$\frac{dI_t}{dt} = \gamma_t E_t - (\mu_t + \sigma_t)I_t \tag{0.7}$$

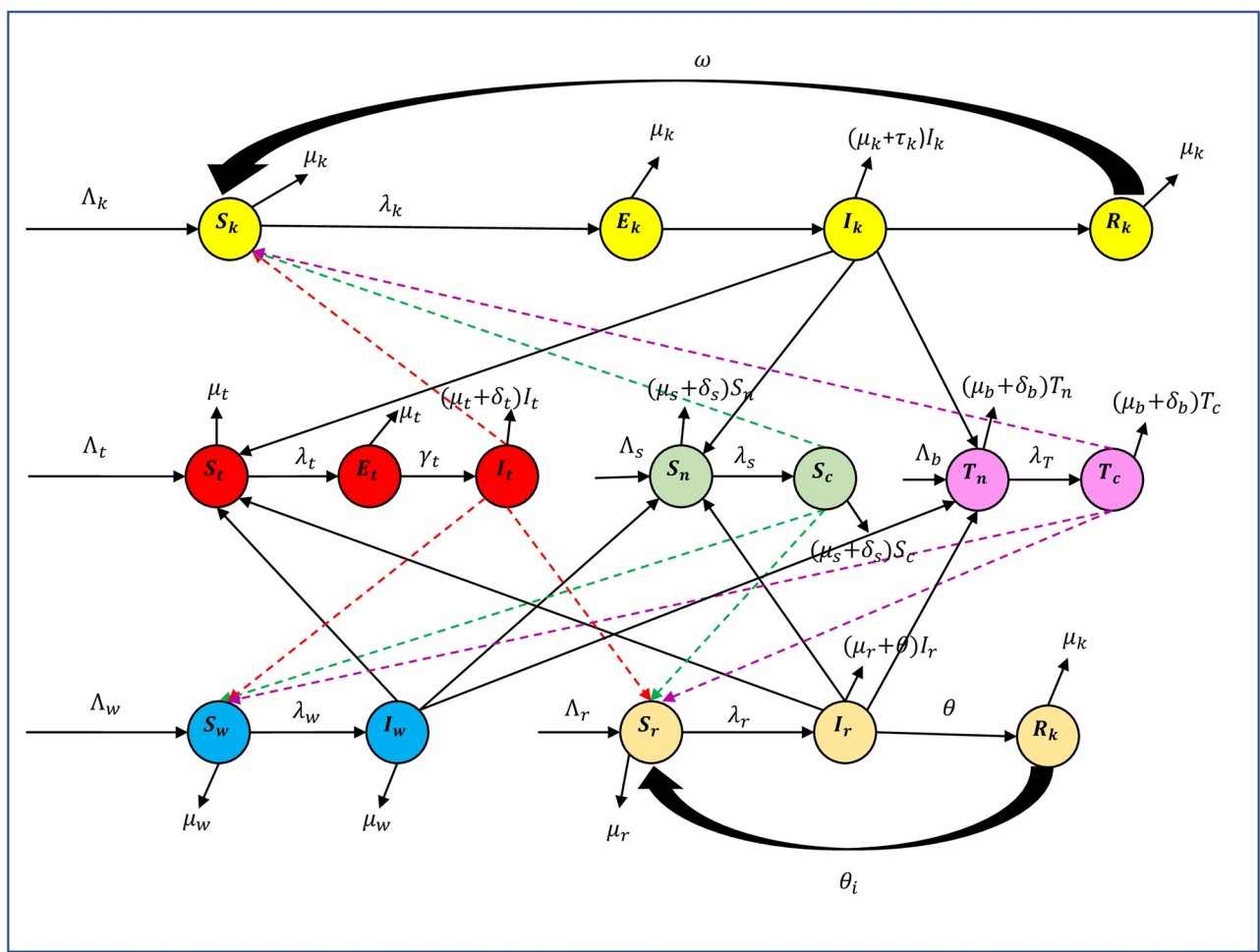

**Fig 1. Compartmental diagram of the constructed model involving the interaction of cattle, small ruminants, wildlife, tsetse flies, tabanids and *Stomoxys* with *Trypanosoma* species.** NB: we denote $\lambda_k = bS_k(\varphi k1 I_t + \varphi_{k2} T_c + \varphi k3 S_c)$, $\lambda_t = b\varphi_t S_t(I_k + I_w + I_r)$, $\lambda_s = b\alpha_2 S_n(I_k + I_w + I_r)$, $\lambda_T = b\alpha_1 T_n(I_k + I_w + I_r)$, $\lambda_w = e^{\omega_1} bS_w(\tau_1 I_t + \tau_2 T_c + \tau_3 S_c)$, $\lambda_r = e^{\omega_2} bS_r(\zeta_1 I_t + \zeta_2 T_c + \zeta_3 S_c)$, $\tau = (\tau_1 + \tau_2 + \tau_3)$, $\zeta = (\zeta_1 + \zeta_2 + \zeta_3)$ in the model diagram and model analysis.

$$\frac{dS_n}{dt} = \Lambda_s - b\alpha_2 S_n(I_k + I_r + Iw) - (\mu_s + \delta_s)S_n \tag{0.8}$$

$$\frac{dS_c}{dt} = b\alpha_2 S_n(I_k + I_r + Iw) - (\mu_s + \delta_s)S_c \tag{0.9}$$

$$\frac{dT_n}{dt} = \Lambda_b - b\alpha_1 T_n(I_k + I_r + Iw) - (\mu_b + \delta_b)T_n \tag{0.10}$$

$$\frac{dT_c}{dt} = b\alpha_1 T_n(I_k + I_r + Iw) - (\mu_b + \delta_b)T_c \tag{0.11}$$

$$\frac{dS_w}{dt} = \Lambda_w - e^{\omega_1} bS_w(\tau_1 I_t + \tau_2 T_c + \tau_3 S_c) - \mu_w S_w \tag{0.12}$$

$$\frac{dI_w}{dt} = e^{\omega_1} bS_w(\tau_1 I_t + \tau_2 T_c + \tau_3 S_c) - \mu_w I_w \qquad (0.13)$$

$$\frac{dS_r}{dt} = \Lambda_r - e^{\omega_2} bS_r(\zeta_1 I_t + \zeta_2 T_c + \zeta_3 S_c) + \theta_1 R_r - \mu_r S_r \qquad (0.14)$$

$$\frac{dI_r}{dt} = e^{\omega_2} bS_r(\zeta_1 I_t + \zeta_2 T_c + \zeta_3 S_c) - (\theta + \phi + \mu_r)I_r \qquad (0.15)$$

$$\frac{dR_r}{dt} = \theta I_r - (\theta_1 + \mu_r)R_r \qquad (0.16)$$

The model (0.1)–(0.16) is simulated using the listed parameters as shown in Tables 1 and 2. The symbols, values and list of references were included. Some values were estimated based on the results from the field data in southwest Nigeria.

Since the model monitors changes in the cattle, small ruminant, wildlife, tsetse fly, tabanid and stomoxyine populations, the variables and the parameters are assumed to be non-negative for all $t \geq 0$, therefore Eqs (0.1)–(0.16) were analysed in a feasible region $\mathscr{R}$ of biological interest.

There are factors that affect the transmission dynamics of AAT and its transmitting vectors including environmental variables (temperature, humidity, rainfall, wind speed), vegetational factors (drought, degradation, deforestation, overcrowding) and conflicts (cattle rustling, insurgencies, social disagreements). Tsetse reproduction, densities, abundance, distribution and susceptibility are indirectly affected by these aforementioned factors [15, 29]. For the intervention strategy using trypanocides, we assumed that treating cattle with a trypanocidal drug has a 75-100% efficacy. Infected cattle are moved to the recovered class, provided they are not exposed to infected transmitting vectors during the trypanocide withdrawal period. This model suggests that areas free of tsetse flies contributed nothing to the $R_0$. Also, the biting flies could be selective in feeding methods, depending on the treatment status of the cattle herd

**Table 2. *Trypanosoma* species variables.**

| Definition | *T. vivax* | *T.congolense* | *T. b.brucei* | Sources |
|---|---|---|---|---|
| IP in tsetse fly | 10 | 20 | 25 | [19] |
| TR from infected tsetse to susceptible cattle | 0.29 | 0.46 | 0.62 | [20] |
| IR of *G. palpalis*- mouth infections only | 0.281 | 0.001 | 0.516 | [31] |
| IR of *G. tachinides*- mouth infections only | 0.155 | 0.042 | 0.127 | [31] |
| IR of *S. calcitrans*- mouth infections only | 0.153 | - | - | [31] |
| IR of *S. niger niger*- mouth infections only | 0.083 | - | - | [31] |
| IR of *Tabanus* species- mouth infections only | 0.184 | - | - | [31] |
| IP of trypanosomes in *G. palpalis* | 12 | 15 | 12 | [19] |
| IP of trypanosomes in *G. tachinoides* | 12 | 15 | 12 | [19] |
| Natural mortality rate in tsetse | 0.212 | 0.212 | 0.212 | [19] |
| Duration of immunity in cattle | 100 | 100 | 50 | [20] |
| IP in cattle | 12 | 15 | 12 | [20] |
| TR of trypanosomes from vertebrate to tsetse | 0.177 | 0.025 | 0.065 | [20] |

IP = Incubation Period, IR = Infectious Rate, TR = Transmission Rate

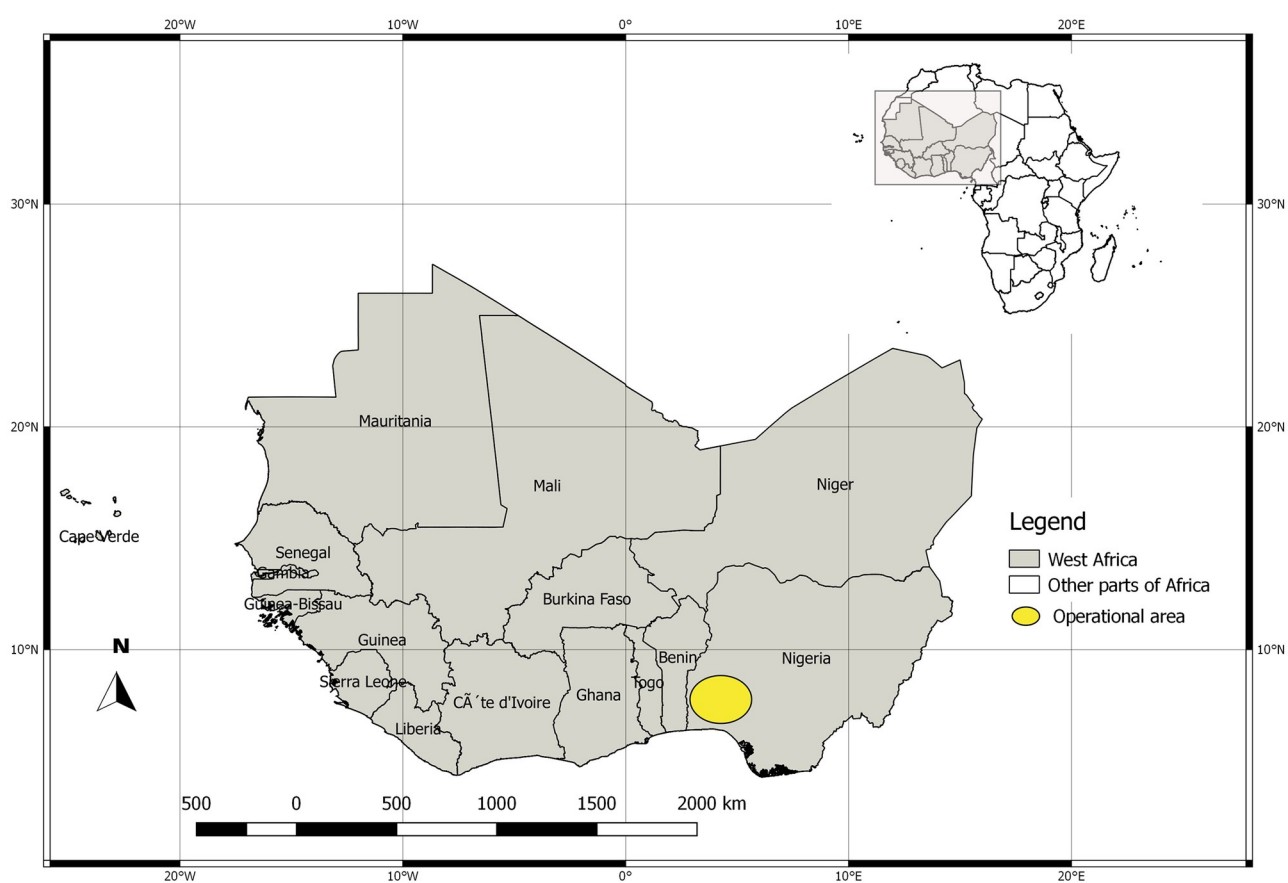

**Fig 2. Map showing West African countries and the operational area in southwest Nigeria.**

with trypanocides and insecticides. The presence of wildlife in the area could change the status of the model especially with some specific species such as *T. vivax*. The strategic application of ITT and ITC on the vector flies using a regional approach concluded the model. This strategic control method is feasible in West African countries (Fig 2).

**Assessment of control strategy in southwest Nigeria using Tsetse Plan.** In order to validate our model, experimental and field assessment was done using Tsetse Plan (available from tsetse.org). The software provides implementation strategy and control cost of AAT directly from the field. This study was carried out in southwest Nigeria between April 2016 and March 2017. The study area landmass is about 78, 000 km$^2$, however, cattle settlement area is about $\frac{1}{1000th}$ of this total area. Blood samples were randomly collected from 745 cattle in southwest Nigeria, and screened for trypanosome DNA [3]. Nzi traps were set in the study sites to effectively trap both tsetse and biting flies [30, 31]. The values from these experiments were computed as parameters in the model. The landing preference of tsetse flies and biting flies on cattle suggests that insecticidal application methods can improve the control strategy [32, 33].

The tsetse software analyses the control plans and their relative costs for AAT in a geographical area (also known as Tsetse Plan). Targeted animals to be protected were livestock (cattle and small ruminants). The trapped tsetse species in the preliminary study were *G. palpalis* and *G. tachinoides* (Palpalis group) which is in line with the stability model. The population density of the tsetse species was assumed to be medium (fly density estimated at 300—1000 flies per km$^2$) from the overall report. A "cross" shape was selected to represent the tsetse

distribution for the affected study area, which indicates that invasion is from the north, east and west (adjacent invasion areas). It was estimated that 10,000 cattle graze in the operational area that are available for insecticidal treatment (ITC), excluding those involved with zero grazing. The small ruminant population was 75% larger than the cattle population (i.e. 17,500 small ruminants) in this location. These livestock were not often targeted during treatment and mostly show subclinical signs of trypanosomosis. However, the insecticidal treatment (ITC) has been frequently used to directly protect livestock against tsetse flies and biting flies. Wildlife is assumed to be absent in a 40 km$^2$ area out of the 70 km$^2$ operational study area in this simulation, which indicates a very low wildlife reservoir (with approximately five wildlife per km$^2$), while cattle do not have contact with wildlife in grazing areas.

## Pertinent data obtained from the project area with tsetse plan

Elimination or continuous control of vector flies and AAT in a hypothetical area of 10,000 km$^2$ located in southwest Nigeria was examined to validate our model. The estimated area that should be completely cleared of tsetse flies from the model (this area needs baits for at least eight months, while clearance occurs) is 1386 km$^2$. However, areas where invading tsetse will occur at much reduced density (this area will need baits indefinitely to deal with invaders) is 378 km$^2$. Hence, the total calculated project area is 1764 km$^2$. In this study, biting flies (mechanical vectors) are also considered to be involved in the transmission of trypanosomes, therefore the choice of trap is important in the project. Since the project area is subject to tsetse invasion, the invading vector flies must enter the area to be killed. There will always be few tsetse in those areas that are less than about 3 km$^2$ from the invasion front. If baits are not maintained properly near the front, the flies will invade in greater numbers, possibly nullifying previous control efforts.

## Expectation from the operational area

Since the flies will not be removed totally and permanently from the project area, the cattle will still need to be examined for trypanosomosis and administered trypanocidal drugs, albeit perhaps less intensively. The vegetation area needs to be considered in the planning since it affects the tsetse species distribution (S1 Fig). Hence, baited Nzi traps should be considered for effectiveness. Cattle in the middle of the operational area, where the flies need be reduced or were absent, might go much nearer to the invasion source(s) to graze and drink, increasing the risk of infection (S2–S4 Figs).

Obvious concerns of increasing the size of the cleared area relative to the invaded area is expected to make invasion less significant. The estimated densities of various sizes of cattle are shown to be: large cattle (> 150 kg)- 4.7 per km$^2$ and small cattle (< 150 kg)- 4.7 per km$^2$. The calculated numbers of bait estimated for the control operations are (i) treated cattle- larger animals only (760), (ii) targets- without odours (11,300), (iii) traps- without odours (250).

## Cost estimation

There are thirteen input stages in preparing the cost estimation with Tsetse Plan software. The cost analysis is technique dependent. In this study, trapping, ITC and ITT were considered in the model as viable control techniques for western Africa. In cases of animal movement where sufficient livestock are present for ITC, insecticides are often applied by spraying (restricted-insecticide application protocol (RAP) has been cost effective) or pour on. The cost of traps is relatively high (not necessarily the cost of acquiring the traps), because of the cost of manpower required for deployment which is dependent on density. Notably, the targeted vector

flies were *Glossina* species and biting flies, which means that a combination of techniques was required.

## Data analysis

Epidemiological data from field studies were entered into the Tsetse Plan software which automatically generates post-analysis results. The estimation of parameters was achieved using the least squares method in Excel solver [34], with a view to minimising summation of squared errors given by $\Sigma(Y(t,p) - X_{real})^2$ subject to the AAT model (0.1)–(0.16) where $X_{real}$ is the field reported data, and $Y(t,p)$ represents the solution of the model corresponding to the number of active cases divided by time t with the set of estimated parameters denoted by $p$. The numerical simulations were conducted using Maple 17 software and the results illustrate the system's behaviour for different values of model parameters. Some of the parameters were estimated from the epidemiological data. Population variables for the model were considered across the study area in western Africa using QGIS software (Version 2.18).

## Results

Population growth of cattle is observed when there is a reduction in the infected cattle population compared to the susceptible cattle population due to the intervention strategies.

The growth can be narrowed if there is a problem with the balance such as problems of trypanocidal and insecticidal resistance (for biting flies, which could pose continuous challenge), changing climate, epidemics from other infectious diseases, ecological instability and human activities.

**Theorem 1**: The feasible region $\mathscr{R}$ defined by
$\{S_k(t), E_k(t), I_k(t), R_k(t), S_t(t), E_t(t), I_t(t), S_n(t), S_c(t), T_n(t), T_c(t), S_w(t), I_w(t), S_r(t), I_r(t), R_r(t) \in \mathscr{R}^{16} : N_k(0) \leq N_k(t) \leq \frac{\Lambda_k}{\mu_k}, N_t(0) \leq N_t(t) \leq \frac{\Lambda_t}{\mu_k}, N_s(0) \leq N_s(t) \leq \frac{\Lambda_s}{\mu_s}, N_b(0) \leq N_b(t) \leq \frac{\Lambda_b}{\mu_b}, N_w(0) \leq N_w(t) \leq \frac{\Lambda_w}{\mu_w}, N_r(0) \leq N_r(t) \leq \frac{\Lambda_r}{\mu_r}\}$ with initial conditions $S_k(0){>}0, E_k(0){\geq}0, I_k(0){\geq}0, R_k(0){\geq}0, S_t(0){\geq}0, E_t(0){\geq}0, I_t(0){\geq}0, S_n(0){\geq}0, S_c(0){\geq}0, T_n(0){\geq}0, T_c(0){\geq}0, S_w(0){\geq}0, I_w(0){\geq}0, S_r(0){\geq}0, I_r(0){\geq}0, R_r(0){\geq}0$ is positive invariant for system (0.1)–(0.16).

**Proof**: If the vertebrate hosts (cattle, small ruminants and wildlife) and invertebrate hosts (tsetse fly, stomoxyines, tabanids) population sizes are given by $N_k(t) = S_k(t) + E_k(t) + I_k(t) + R_k(t)$, $N_t(t) = S_t(t) + E_t + I_t$, $N_s(t) = S_n(t) + S_c$, $N_b(t) = T_n(t) + T_c$, $N_w(t) = S_w(t) + I_w(t)$ and $N_r(t) = S_r(t) + I_r(t) + R_r(t)$. Adding the first four equations of the model (0.1)–(0.16) gives $\frac{dN_k}{dt} \leq \Lambda_k - \mu_k N_k(t)$ so that $N_k(t) \to \frac{\Lambda_k}{\mu_k}$ as $t \to \infty$. Thus $\frac{\Lambda_k}{\mu_k}$ is an upper bound of $N_k(t)$ provided that $N_k(0) \leq \frac{\Lambda_k}{\mu_k}$. Further, if $N_k(0) > \frac{\Lambda_k}{\mu_k}$ then $N_k(t)$ will decrease to this level. Similar calculation for Eqs (0.5)–(0.13) and (0.14)–(0.16) shows that, $N_t \to \frac{\Lambda_t}{\mu_t}, N_s \to \frac{\Lambda_s}{\mu_s}$ and $N_t \to \frac{\Lambda_s}{\mu_s}, N_w \to \frac{\Lambda_w}{\mu_w}$ and $N_r \to \frac{\Lambda_r}{\mu_r}$, respectively, as $t \to \infty$. Thus, the following feasible region:

$$\mathscr{R} = \{S_k(t), E_k(t), I_k(t), R_k(t), S_t, E_t, I_t, S_n, S_c, T_n, T_c, S_w, I_w, S_r, I_r, R_r \in \mathscr{R}^{16} : N_k(t) \leq \frac{\Lambda_k}{\mu_k},$$

$$N_t(t) \leq \frac{\Lambda_t}{\mu_t}, N_s(t) \leq \frac{\Lambda_s}{\mu_s}, N_b(t) \leq \frac{\Lambda_b}{\mu_b}, N_w(t) \leq \frac{\Lambda_w}{\mu_w}, N_r(t) \leq \frac{\Lambda_r}{\mu_r}\}$$

is positively- invariant. For the full proof to theorem 1 see Appendix A.

## Disease-free equilibrium point

For the disease-free equilibrium, the disease states and the left-hand side of (0.1)–(0.16) were set to zero. The resulting system is solved which is given

$$\pi_0 = \left( \frac{\Lambda_k}{\mu_k}, 0, 0, 0, \frac{\Lambda_t}{\mu_t}, 0, 0, \frac{\Lambda_s}{\mu_s}, 0, \frac{\Lambda_b}{\mu_b}, 0, \frac{\Lambda_w}{\mu_w}, 0, \frac{\Lambda_r}{\mu_r}, 0, 0 \right)$$

For the ITC method of insecticidal control, vectors were targeted on the cattle rather than the parasite (trypanosomes), hence, there is mortality at the point of feeding and those occurring between feeds for all the vector flies. The probability of the vector fly surviving a feed is therefore, the product of the probabilities it feeds on untreated hosts and feeds on treated hosts and survives that meal. Therefore, we assumed that vector flies feed off all cattle at random, with respect to the cattle treatment status. Due to pastoralism and nomadic management systems being widely practised in West Africa, ITC is very effective because livestock farmers move their animals around for grazing. For ITT, areas beyond the target animals were also considered, as the vector fly populations of adjacent areas were also targeted. This is an effective method in elimination strategy, it also targets flies within and outside the target areas. ITT is most effective in areas where zero-grazing is practised. The use of ITC was more effective than ITT in the West Africa model, in which the feeding tsetse and other biting flies are affected largely because of management practises. To obtain $R_0$ for the the model (0.1)–(0.16), the next generation matrix technique earlier described by Diekmann et al [35], which was further reviewed by Van den Driessche and Watmough [36] was utilised.

$FV^{-1}$ is called the next generation matrix.

$$\mathbf{F} = \left[ \frac{\partial \mathcal{F}_i}{\partial x_i} (\bar{x}) \right] \quad \text{and} \quad \mathbf{V} = \left[ \frac{\partial \mathcal{V}_i}{\partial x_i} (\bar{x}) \right].$$

Therefore, the basic reproduction number, $R_0$, is given by

$$R_0 = \rho(\mathbf{FV}^{-1})$$

where $\rho$ is the spectral radius of the product, $\mathbf{FV^{-1}}$ (i.e, the dominant eigenvalue of $\mathbf{FV^{-}1}$), known as the next generation matrix.

Applying this technique to model (0.1)–(0.16), we let $x = (E_k(t), I_k(t), E_t(t), I_t(t), S_c(t), T_c(t), I_w(t), I_r(t))^T$. Then model (0.1)–(0.16) can be written as $\frac{dx}{dt} = \mathcal{F}(x) - \mathcal{V}(x)$, where, the rate of new appearance of new infection in the compartments of the model (0.1)–(0.16) is obtained as

$$\mathcal{F}(x) = \begin{pmatrix} bS_k(\varphi_{k1}I_t + \varphi_{k2}T_c + \varphi_{k3} \\ 0 \\ b\varphi_t S_t(I_k + I_r + I_w) \\ 0 \\ b\alpha_2 S_n(I_k + I_r + I_w) \\ b\alpha_1 T_n(I_k + I_r + I_w) \\ e^{\omega_1} bS_w(\tau_1 I_t + \tau_2 T_c + \tau_3 S_c) \\ e^{\omega_2} bS_r(\zeta_1 I_t + \zeta_2 T_c + \zeta_3 S_c) \end{pmatrix}$$

and the rate of individuals into and out of the compartments of the model (0.1)–(0.16) are defined as

$$
\mathcal{V}(x) = \begin{pmatrix}
(\alpha_k + \mu_k)E_k \\
(\rho r_k + \tau_k + \mu_k)I_k - \alpha_k E_k \\
(\gamma_t + \mu_t + \sigma_t)E_t \\
(\mu_t + \sigma_t)I_t - \gamma_t E_t \\
(\mu_s + \delta_s)S_c \\
(\mu_b + \delta_b)T_c \\
\mu_w I_w \\
(\theta + \varphi + \mu_r)I_r
\end{pmatrix}
$$

Find the derivative of $\mathcal{F}(x)$ and $\mathcal{V}(x)$ at the disease-free equilibrium point $\pi_0 = \left(\frac{\Lambda_k}{\mu_k}, 0, 0, 0, \frac{\Lambda_t}{\mu_t}, 0, 0, \frac{\Lambda_s}{\mu_s}, 0, \frac{\Lambda_b}{\mu_b}, 0, \frac{\Lambda_w}{\mu_w}, 0, \frac{\Lambda_r}{\mu_r}, 0, 0\right)$ with respect to the disease classes results into **F** and **V** respectively, where,

$$
\mathbf{F} = \begin{pmatrix}
0 & 0 & 0 & \frac{b\varphi_{k1}\Lambda_k}{\mu_k} & \frac{b\varphi_{k3}\Lambda_s}{\mu_s} & \frac{b\varphi_{k2}\Lambda_b}{\mu_b} & 0 & 0 \\
0 & 0 & 0 & 0 & 0 & 0 & 0 & 0 \\
0 & \frac{b\varphi_t\Lambda_t}{\mu_t} & 0 & 0 & 0 & 0 & \frac{b\varphi_t\Lambda_t}{\mu_t} & \frac{b\varphi_t\Lambda_t}{\mu_t} \\
0 & 0 & 0 & 0 & 0 & 0 & 0 & 0 \\
0 & \frac{b\alpha_2\Lambda_s}{\mu_s} & 0 & 0 & 0 & 0 & \frac{b\alpha_2\Lambda_s}{\mu_s} & \frac{b\alpha_2\Lambda_s}{\mu_s} \\
0 & \frac{b\alpha_1\Lambda_b}{\mu_b} & 0 & 0 & 0 & 0 & \frac{b\alpha_1\Lambda_b}{\mu_b} & \frac{b\alpha_1\Lambda_b}{\mu_b} \\
0 & 0 & 0 & \frac{e^{\omega_1}\tau_1\Lambda_w}{\mu_w} & \frac{e^{\omega_1}\tau_3\Lambda_w}{\mu_w} & \frac{e^{\omega_1}\tau_2\Lambda_w}{\mu_w} & 0 & 0 \\
0 & 0 & 0 & \frac{e^{\omega_2}\zeta_1\Lambda_w}{\mu_w} & \frac{e^{\omega_2}\zeta_3\Lambda_r}{\mu_r} & \frac{e^{\omega_2}\zeta_2\Lambda_r}{\mu_r} & 0 & 0
\end{pmatrix}
$$

$$
\mathbf{V} = \begin{pmatrix}
(\alpha_k + \mu_k) & 0 & 0 & 0 & 0 & 0 & 0 & 0 \\
-\alpha_k & \rho r_k + \tau_k + \mu_k & 0 & 0 & 0 & 0 & 0 & 0 \\
0 & 0 & \gamma_t + \mu_t + \sigma_t & 0 & 0 & 0 & 0 & 0 \\
0 & 0 & -\gamma_t & \mu_t + \sigma_t & 0 & 0 & 0 & 0 \\
0 & 0 & 0 & 0 & \mu_s + \delta_s & 0 & 0 & 0 \\
0 & 0 & 0 & 0 & 0 & \mu_b + \delta_b & 0 & 0 \\
0 & 0 & 0 & 0 & 0 & 0 & \mu_w & 0 \\
0 & 0 & 0 & 0 & 0 & 0 & 0 & \theta + \phi + \mu_r
\end{pmatrix}
$$

since **V** is a non-singular matrix, the inverse of **V**, **V⁻¹** can be obtained as

$$
\mathbf{V^{-1}} =
\begin{pmatrix}
\frac{1}{(\alpha_k+\mu_k)} & 0 & 0 & 0 & 0 & 0 & 0 & 0 \\
\frac{\alpha_k}{(\alpha_k+\mu_k)(\rho r_k+\tau_k+\mu_k)} & \frac{1}{(\rho r_k+\tau_k+\mu_k)} & 0 & 0 & 0 & 0 & 0 & 0 \\
0 & 0 & \frac{1}{(\gamma_t+\mu_t+\sigma_t)} & 0 & 0 & 0 & 0 & 0 \\
0 & 0 & \frac{\gamma_t}{(\gamma_t+\mu_t+\sigma_t)(\mu_t+\sigma_t)} & 0 & 0 & 0 & 0 & 0 \\
0 & 0 & 0 & \frac{1}{\mu_t+\delta_t} & \frac{1}{\mu_s+\delta_s} & 0 & 0 & 0 \\
0 & 0 & 0 & 0 & 0 & \frac{1}{\mu_b+\mu_b} & 0 & 0 \\
0 & 0 & 0 & 0 & 0 & 0 & \frac{1}{\mu_w} & 0 \\
0 & 0 & 0 & 0 & 0 & 0 & 0 & \frac{1}{\theta+\phi+\mu_r}
\end{pmatrix}
$$

The basic reproduction number $R_{01} = \rho(\mathbf{FV^{-1}})$, is the spectral radius of the product $\mathbf{FV^{-1}}$. Hence, for the model (0.1)–(0.16), we arrive at

$$
R_{01} = \sqrt{e^{\omega_1+\omega_2}\left[\frac{b^2\alpha_k\varphi\tau\zeta\varphi_t\Lambda_k\Lambda_r\Lambda_w\Lambda_t\Lambda_s\Lambda_t\alpha_t\alpha_1\alpha_2}{\mu_k\mu_t\mu_s\mu_b\mu_w\mu_r(\alpha_k+\mu_k)(\rho r_k+\tau_k+\mu_k)(\gamma_t+\sigma_t+\mu_t)(\sigma_t+\mu_t)(\mu_b+\delta_b)(\mu_s+\mu_s)(\theta+\phi+\mu-r)}\right]}
$$

In areas where tsetse flies were absent (*T. vivax* showed $R_{02} > 1$ in the cattle population, when infected wildlife hosts are present), and biting flies (tabanids and stomoxyines) are abundant, the basic reproduction number of AAT increases.

$$
R_{02} = e^{\omega_1+\omega_2}\left[\frac{b^2\alpha_k\Lambda_k\tau\zeta\Lambda_k\Lambda_w\Lambda_s\Lambda_b}{\mu_k\mu_w\mu_s\mu_b(\alpha_k+\mu_k)(\rho r_k+\tau_k+\mu_k)(\mu_b+\delta_b)(\mu_s+\mu_s)}\right]
$$

Where $\omega_1$ is the growth rate of infected wildlife introduced in the susceptible population and $\omega_2$ is the growth rate of infected small ruminants introduced in the susceptible population. It is assumed that tsetse flies can also become infected when feeding on infected wildlife. This is common for the Morsitans and Palpalis groups which are present in West Africa. Infected tsetse can then infect susceptible hosts while taking a bloodmeal if the trypanosome infection matures. The field-data from southwest Nigeria entered for the mathematical model generated valuable results applicable to West African countries (Fig 2).

## Global stability of disease-free equilibrium

The global asymptotic stability of the AAT-free equilibrium for the special case with no loss of immunity acquired by the recovered cattle and small ruminants after the treatment of trypanocides was evaluated using a similar approach [37, 38]. With the variables in the model, *T. congolense* and *T. vivax* can be controlled if all the cattle in the herd are continuously treated with trypanocides. Although, *T. brucei* infection showed the lowest challenge, and could even be eliminated provided the sources of infection are only from cattle. The possibility of continuous trypanocide treatment in the presence of infected vector flies poses trypanocide resistance risk to the cattle herd. AAT control strategy in the flock where there were presence of tsetse flies, tabanids and stomoxyines involved the continuous use of insecticides. Proportionate

percentage of the transmitting vectors are thought to harbour *T. congolense*, *T. vivax* and *T. b. brucei*. In follow-up to global stability analyses [39–41], we have the following results.

**Theorem 2**: The disease-free equilibrium, $\pi_0$, of the model (0.1)–(0.16), is globally asymptotically stable in $\mathscr{R}$ if $R_{01} \leq 1$.

**Proof**: Considering the system's Lyapunov function,

$$F = \frac{\alpha_k E_k(t)}{\mu_t \mu_b \mu_s (\alpha_k + \mu_k)(\rho r_k + \tau_k + \mu_k)} + \frac{I_k(t)}{\mu_t \mu_b \mu_s (r_h(a_i) + \tau_k + \mu_k)} + \frac{E_t(t)}{b \tau \Lambda_t} + \frac{I_t(t)(\gamma_t + \sigma_t + \mu_t)}{b \gamma_t \Lambda_t}$$
$$+ \frac{S_c(t)(\mu_s + \delta_s)}{b \Lambda_s} + \frac{T_c(t)(\mu_b + \delta_b)}{b \Lambda_b} + \frac{I_w(t)}{\mu_w} + \frac{I_r(t)}{\theta + \phi + \mu_r} \tag{0.17}$$

we take the time derivative of Eq (0.17) along the solutions of the model (0.1)–(0.17) and simplify to achieve the following bounds:

$$\dot{F} \leq 0 \quad for \quad R_{01} \leq 1$$

$$\dot{F} = 0 \quad if \quad and \quad only \quad I_t(t) = 0$$

$\pi_0$ is globally asymptotically stable in $R$ if $R_{01} \leq 1$. The full proof to theorem 2 is presented in appendix B.

## Global stability of endemic equilibrium

In this section, we investigate the global stability of endemic equilibrium of the model (0.1)–(0.16).

In this scenario, "tsetse-free" areas could have an abundance of biting flies (tabanids and stomoxyines) which could be contaminated with *T. vivax* from infected groups of wildlife species, small ruminants or infected cattle outside the treated zone. For instance, *T. vivax* has managed to maintain itself in South America where tsetse flies are absent, with Tabanidae and *Stomoxys* acting as mechanical vectors [42]. Similarly, Anene et al. [7] observed that *T. vivax* was maintained in the flock by tabanids in tsetse-free areas in Nigeria.

**Theorem 3**: The unique endemic equilibrium, $E_e^*$, of the model (0.1)–(0.16) is globally asymptotically stable if $R_{02} > 1$.

**Proof**: Let $R_{01} > 1$ and $R_{02} > 1$ so that a unique endemic equilibrium exists and consider the following nonlinear Lyapunov function defined by

$$V = S_k(t) - S_k^* - S_k^* \ln\left(\frac{S_h(t)}{S_k^*}\right) + E_k(t) - E_k^* - E_k^* \ln\left(\frac{E_k(t)}{E_k^{**}}\right) + \frac{(\rho r_k + \tau_k + \mu_k)}{\tau_k}\left[I_k(t) - I_k^* - I_k^* \ln\left(\frac{I_k(t)}{I_k^*}\right)\right]$$
$$+ S_t - S_t^* - S_t^{**} \ln\left(\frac{S_t}{S_m^*}\right) + E_t - E_t^* - E_t^* \ln\left(\frac{E_t}{E_t^*}\right) + \frac{(\sigma_t + \mu_t)}{\sigma_t}\left[I_t - I_t^* - I_t^* \ln\left(\frac{I_t}{I_t^*}\right)\right]$$
$$+ S_n - S_n^* - S_n^{**} \ln\left(\frac{S_n}{S_n^*}\right) + \frac{(\mu_s + \delta_s)}{\delta_s}\left[S_c - S_c^* - S_c^* \ln\left(\frac{S_c}{S_c^*}\right)\right] + T_n - T_n^* - T_n^{**} \ln\left(\frac{T_n}{T_n^*}\right)$$
$$+ \frac{(\mu_b + \delta_b)}{\delta_b}\left[T_c - T_c^* - T_c^* \ln\left(\frac{T_c}{T_c^*}\right)\right] + \frac{1}{\mu_w}\left[I_w(t) - I_w^* - I_w^* \ln\left(\frac{I_w(t)}{I_w^*}\right)\right] + \frac{\theta + \phi + \mu_r}{\phi}\left[I_r(t) - I_r^* - I_r^* \ln\left(\frac{I_r(t)}{I_r^*}\right)\right] \tag{0.18}$$

we take the derivative of Eq (0.18) along the solutions of the model (0.1)–(0.16) and simplify to

achieve the following bounds

$$\dot{V} = -V_1 - V_2 - b\varphi S_k^*(I_t^* + T_c^* + S_c^*)\left[1 - \frac{I_t T_c S_c}{I_t^* T_c^* S_c^*} + \frac{I_k(t)}{I_k^*} - \frac{I_k(t)I_t^* T_c^* S_c^*}{I_k^* I_t T_c S_c}\right]$$

$$-V_3 - V_4 - e^{\omega_1} b\tau S_w(I_t + T_c + S_c)\left[1 - \frac{I_t T_c S_c}{I_t^* T_c^* S_c^*} + \frac{I_w(t)}{I_w^*} - \frac{I_w(t)I_t^* T_c^* S_c^*}{I_w^* I_t T_c S_c}\right]$$

$$-v_5 - V_6 - e^{\omega_2} b\zeta S_r(I_t + T_c + S_c)\left[1 - \frac{I_t T_c S_c}{I_t^* T_c^* S_c^*} + \frac{I_r(t)}{I_r^*} - \frac{I_r(t)I_t^* T_c^* S_c^*}{I_r^* I_t T_c S_c}\right]$$

$$-V_7 - V_8 - b\varphi_t S_t(I_k + I_w + I_r)\left[1 - \frac{I_k I_w I_r}{I_k^* I_w^* I_r^*} + \frac{I_t T_c S_c}{I_t^* T_c^* S_c^*} - \frac{I_t T_c S_c I_k^* I_w^* I_r^*}{I_t^* T_c^* S_c^* I_k I_w I_r}\right]$$

$$-V_9 - V_{10} - b\alpha_1 T_n(I_k + I_w + I_r)\left[1 - \frac{I_k I_w I_r}{I_k^* I_w^* I_r^*} + \frac{I_t T_c S_c}{I_t^* T_c^* S_c^*} - \frac{I_t T_c S_c I_k^* I_w^* I_r^*}{I_t^* T_c^* S_c^* I_k I_w I_r}\right]$$

$$-V_{11} - V12 - b\alpha_2 S_n(I_k + I_w + I_r)\left[1 - \frac{I_k I_w I_r}{I_k^* I_w^* I_r^*} + \frac{I_t T_c S_c}{I_t^* T_c^* S_c^*} - \frac{I_t T_c S_c I_k^* I_w^* I_r^*}{I_t^* T_c^* S_c^* I_k I_w I_r}\right]$$

(0.19)

further algebraic manipulation gives

$$f(x, y, z) = x + \frac{y}{x} + \frac{z}{y} + \frac{1}{z} - 4$$

(0.20)

It is sufficient to show that $f(x, y, z) \geq 0$. Since $f_x = f_y = f_z = 0$ gives rise to $x = y = z$ and that $f_{xx} > 0, f_{yy} > 0, f_{zz} > 0$, one see that the minimum of $f(x, y, z)$ is attainable at $x = y = z$. In what follows, (Eq 0.20) is reduced to $(x - 1)^2 \geq 0$ or $(y - 1)^2 \geq 0$ or $(z - 1)^2 \geq 0$ with equality if and only if $x = 1$ or $y = 1$ or $z = 1$ respectively. Hence, $V_2 \geq 0$. The proof of $V_3 \geq 0$ is similar to $V_1 \geq 0$ while that of $V_4 \geq 0$ is similar to $V_2 \geq 0$ and so on. Whenever $\frac{I_k^*}{I_k(t)} = \frac{I_t^*}{I_t} = \frac{T_c^*}{T_c} = \frac{S_c^*}{S_c}$, $\frac{I_w^*}{I_w(t)} = \frac{I_t^*}{I_t} = \frac{T_c^*}{T_c} = \frac{S_c^*}{S_c}, \frac{I_r^*}{I_r(t)} = \frac{I_t^*}{I_t} = \frac{T_c^*}{T_c} = \frac{S_c^*}{S_c}$ it follows from (0.19) that $\dot{V} \leq 0$ with $\dot{V} = 0$ if and only if $S_k(t) = S_k^*(t), E_k(t) = E_k^*, I_k(t) = I_k^*, S_t = S_t^*, E_t = E_t^*, I_t = I_t^*, S_n = S_n^*, S_c = S_c^*, T_n = T_n^*, T_c = T_c^* S_w = S_w^*, S_r = s_r^*, I_w = I_w^*, I_r = I_r^*$. This further implies that $R_k(t) = \frac{\rho r_k I_k^*}{\mu_k} = R_k^*, \frac{\theta R_r^*}{\mu_r} = R_r^*$ since $(S_k(t), E_k(t), I_k(t), S_w, I_w, S_r, I_r)$ tends to $(S_t^*, E_t^*, I_t^*, S_n^*, S_c^*), T_n^*, T_c^*$ as $t \to \infty$. Therefore, LaSalle's principle explains that the largest compact invariant subset of the set where $\dot{V} = 0$ is the endemic equilibrium point $E_e^*$ [43]. Hence, every solution in $\mathscr{R}$ approaches $E_e^*$ for $R_{01}, R_{02} > 1$, and $E_e^*$ is globally asymptotically stable. For the full proof to theorem 3 see Appendix C.

## Local stability of disease-free and endemic equilibrum

**Theorem 4**: The disease-free equilibrium point, $\pi_0$, is locally asymptotically stable (LAS) if $R_{01} < 1$ and unstable if $R_{01} > 1$.

**Theorem 5**: The trypanosomosis model (0.1)–(0.16) has a unique endemic equilibrium whenever $R_{01}, R_{02} > 1$.

**Remarks**: It is worth mentioning that local stability of an equilibrium (situation) would imply existence of that situation for a short time (depending on certain circumstances or conditions). Whereas global stability would imply existence of a situation forever regardless of any condition. In summary, global stability of system (model) implies local stability, however, local stability of system does not imply global stability.

## Numerical results

We illustrated the theoretical results established in this study and by considering initial conditions $S_k(0) = 100$, $E_k(0) = 10$, $I_k(0) = 5$, $R_k(0) = 0$, $S_t(0) = 1000$, $E_t(0) = 30$, $I_t(0) = 30$, $S_n(0) = 100$, $S_c(0) = 10$, $T_n(0) = 100$ and $T_c(0) = 10$.

The graph-based behaviour of the cattle populations demonstrate that the susceptible cattle population reduced when infected tsetse flies, contaminated tabanids and stomoxyines, as well as wildlife populations, were present (Fig 3A). The size of the exposed cattle population decreases with progression to the infected group (Fig 3A). The decrease in the number of infected cattle contributes to the increase in number of recovered cattle (Fig 3B).

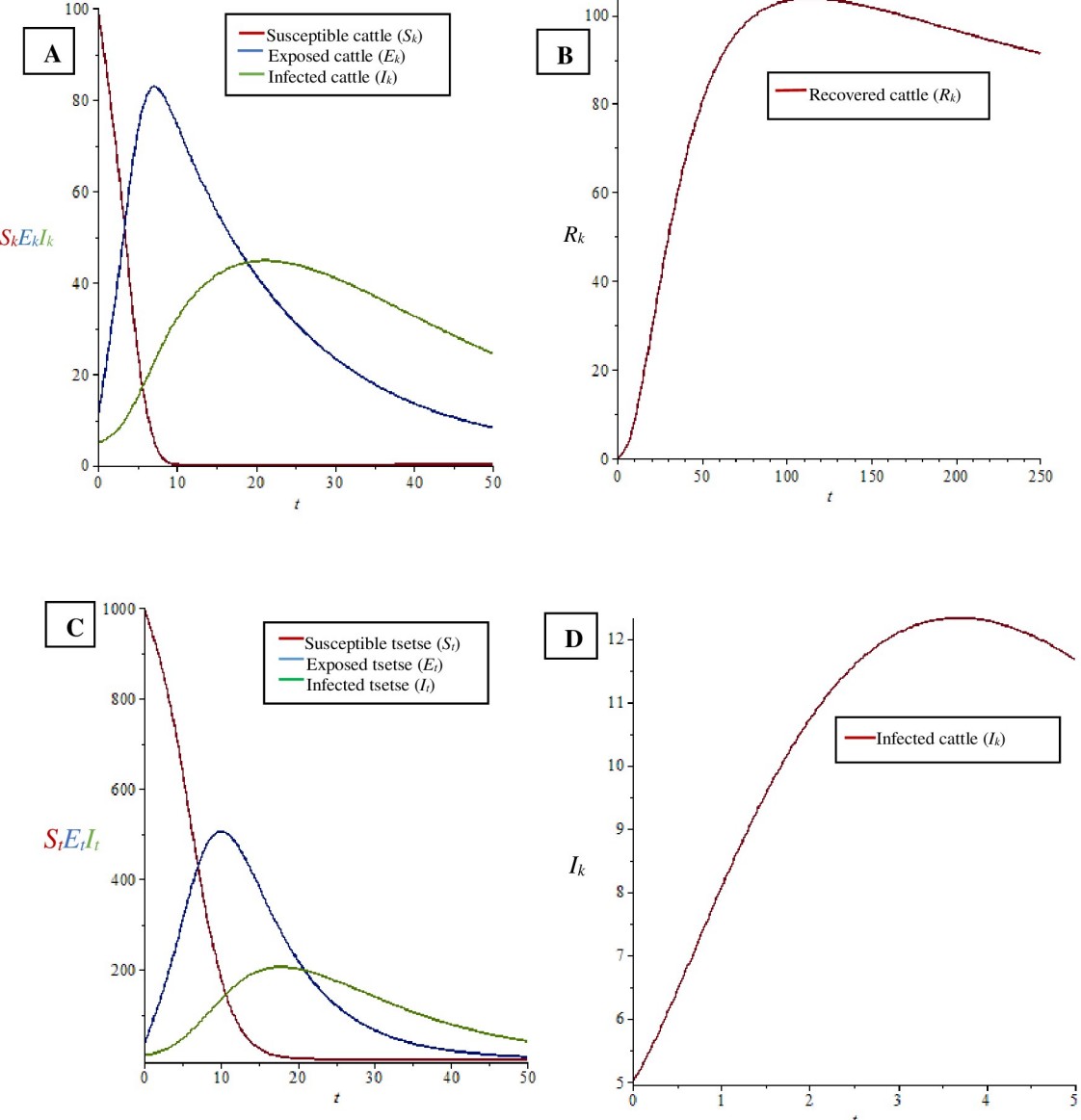

**Fig 3. Behavioural conditions in different populations.** A. The behaviour of cattle population when $R_{01} < 1$. B. The behaviour of recovered cattle when $R_{01} < 1$. C. The behaviour of tsetse fly population when $R_{01} < 1$. D. The behaviour of infected cattle when the tabanid and stomoxyine population increase with abundant presence of wildlife populations.

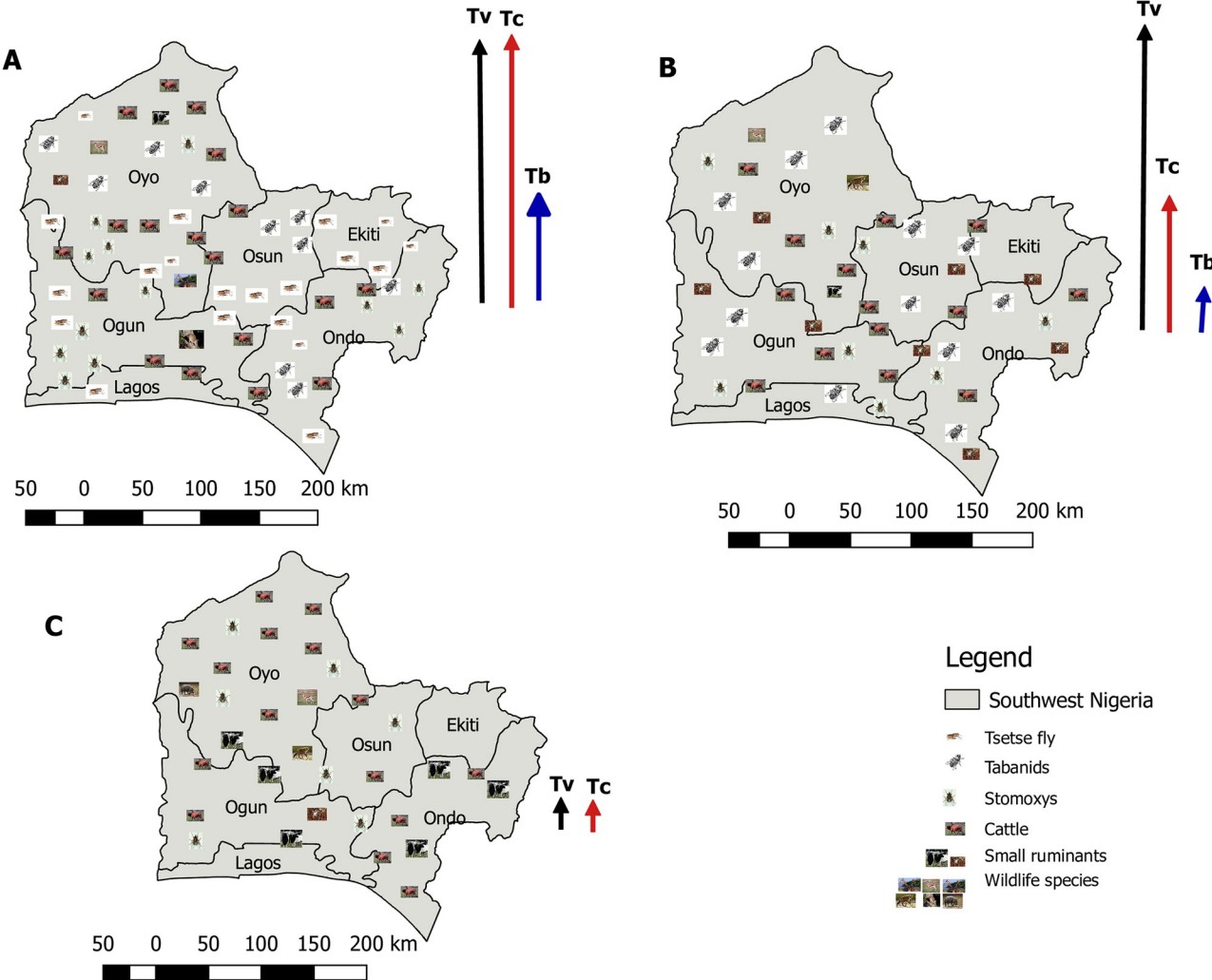

**Fig 4. Simulated situation of AAT in southwest Nigeria.** A. Areas with abundant tsetse flies, biting flies, cattle, small ruminants and wildlife, with a high prevalence of *Trypanosoma* spp. B. Absence of tsetse flies was depicted, but biting flies, cattle, small ruminants and wildlife present, the *T. vivax* is high. C. Absence of all the vectors when ITC and ITT is used along with trypanocides. There are still wildlife reservoirs and herds of cattle and small ruminants. Abbreiations: Tv- *Trypanosoma vivax*, Tc- *Trypanosoma congolense*, Tb- *Trypanosoma brucei brucei*

The illustrated graph of the tsetse fly populations showed that the magnitude of susceptible tsetse fly decreases as a result of infection from infected cattle and the use of insecticide (Fig 3C). The magnitude of the exposed tsetse fly population decreases when they progressed to the infected group and as a result of the use of insecticide (Fig 3C). Finally, the population of infected tsetse flies reduced as a result of the use of insecticide (Fig 3C). A similar relationship holds for tabanid and stomoxyine populations. Cattle infected with *T. vivax* increase when the tabanid and stomoxyine populations increase with abundant presence of wildlife populations even when tsetse are absent (Fig 3D).

In cases where all the factors (cattle, wildlife, small ruminants, tsetse flies and biting flies) are present, there will be high prevalence of *T. vivax, T. congolense* and *T. brucei* (Fig 4A). The use of ITC and trypanocides could eliminate *T. brucei* completely, while *T. vivax* and *T. congolense* persists (Fig 4B). The presence of biting flies only could help increase *T. vivax* significantly in the cattle herd. The wildlife are expected to be left to maintain the ecosystem, while

the presence of small ruminants could serve as reservoirs in the domestic cycle (Fig 4C). Even though some of them remain infected (wildlife and small ruminants), the absence of the transmitting vectors will help with a significant success in the elimination strategy.

## Field-reality model in southwest Nigeria

The field results showed that only 3% (approximately 585,000) of the national cattle population are domiciled in livestock farms in southwest Nigeria. Other livestock (abattoir and trade) are transported from the northern parts of the country. The results of the trypanosome DNA in cattle blood showed a prevalence of 23.8% with highest prevalence of *T. vivax* (13.0%) in the overall study [3]. The study reported highest prevalence of *T. congolense* (11.0%; 95%CI: 8.5-14.2) and *T. vivax* (14.7%; 95%CI: 11.0-19.5) in the wet and dry seasons, respectively. Further studies reported trapped vector flies with densities highest for stomoxyines, followed by tsetse flies and then tabanids [44]. Also, *Trypanosoma* species DNA were found in both the biological and mechanical vector flies with *T. vivax* prominent in the biting flies mechanical vector [31]. All these flies had their bloodmeals from either humans, cattle or wildlife. The wildlife observed from the bloodmeals of tsetse and biting flies included giraffe, hippopotamus, gazelle, spotted hyena and long-tailed rat [32]. However, the absence of *T. b. gambiense* in both cattle and humans in the study sites fits our predictive model. Meanwhile, bloodmeals from small ruminant sources were not detected. Hence, the stability model considers the vertebrate hosts (cattle, small ruminants and wildlife) based on their importance and the transmitting vectors (tsetse flies, tabanids and stomoxyines). A total of 93.9% (95% CI: 88.58-96.92) of livestock owners use trypanocides, while 60.5% (95% CI: 51.84-68.48) use insecticides (ITC only) without specific regimen [17]. Assessment of commonly used insecticides in the cattle herds showed improved results with restricted insecticidal application protocol (RAP)- in which insecticide application was limited to legs and belly) compared to the Fulani application approach (FAA)- insecticide application was applied based on the knowledge of farmers) [33]. Therefore, an elimination approach needs to consider improved insecticides (ITT) in both operational and adjacent areas in an integrated strategy to control AAT.

## Control cost and implementation strategy

We estimated the cost for elimination of an isolated study area with Palpalis group of tsetse flies and biting flies to be between 599—1875 US$ / km$^2$. However, to maintain barriers against reinvasion for the next five years, the cost could increase by 20—50%. However, if barriers are extended to larger areas for a longer period, the cost would increase further. The treatment of cattle in protected areas would be between 4—12 times, depending on the severity of the fly challenge from *Glossina* species and biting flies (S1 and S2 Tables). The price of trypanocides was estimated at US$ 1.67 for > 150 kg adult dose and delivery cost at US$ 5.56, bringing the cost per dose at US$ 0.036 / kg and thus US$ 28.9 per cattle / annum if administered quarterly. The cost could increase to US$ 115.6 / cattle / annum if trypanocides are administered monthly. Hence, if targeted areas are well-protected (traps, ITC, ITT), the trypanocide treatment would only be quarterly. The grand total of the project cost on vector fly suppression per annum was US$ 1,056,990.00, while US$ 16,906,500 would be spent on trypanocides to implement the strategic control of AAT in southwest Nigeria. The overall elimination costs was US$ 17,963,490 per annum in an area of 78,000 km$^2$ of southwest Nigeria, in which elimination could be achieved within three years of consistent control measures. The total cost for both insecticides and trap cloths with miscellaneous cost was estimated at US$ 23,160; provided the traps were made locally to save cost. The cost of traps was US$ 2,640 at local cost (4 traps per km$^2$), US$ 350 for ITC and US$ 23,160 for ITT. Contingency was 10% of total cost, while

facility, administration and staff recruitment for the experiment covers an estimated 88% of the total cost. The overhead cost could increase considering other factors like workshops for local livestock owners and foreign exchange policies. Here, a 10% margin for error produced the results. Expendables for eliminating and monitoring baits were also reported (S1 and S2 Tables). To improve the success rate, the insecticidal approach (ITT) must be continuously maintained both within and outside the operational area. Barriers between the domestic and sylvatic areas need to be made active, while periodical assessment of cattle blood should be a routine practice. In the presence of vector flies, quarterly use of trypanocides and ITC on ruminants in a structured manner across the study areas need to be instituted.

## Discussion

In this model, mathematical tools for investigating the conditions for control of AAT in western Africa were provided. Validation of the model was supported from field data and Tsetse Plan control methods. There was general improvement on previous models which considered tsetse flies (biological agent) as the only transmitting vector agent of trypanosomiasis. In fact, Rogers' model expresses limitations as it does not consider biting flies in a field-based model because its importance relative to cyclical transmission has not been fully established in the field, which thereafter remained prototype for subsequent models [19, 27]. Notably, advances in mathematical biology over the years made selection of important variables possible and helped introduced several factors and hosts into a single model. Previous limitations on model formulations in which treatment of AAT in cattle herd was limited to trypanocides without considering insecticides and invasion from off-target infected areas [20], were addressed in this model. Importantly, the epidemiological data from southwest Nigeria were entered into the Tsetse Plan software to validate our constructed model for western Africa.

The random feeding of tsetse flies on a given host described earlier [19] was maintained in this model. Inclusion of a random feeding or interrupted feeding mechanism of biting flies was initiated in this model. The tabanids feeding were clearly female, while stomoxyines and tsetse flies involved both sexes. This study showed that the use of trypanocides coupled with ITC and ITT could help to control AAT in livestock herds. However, the model considers some areas in western Africa which had few or no tsetse flies present in which trypanocides are only given when cattle show signs of disease. Besides, either ITC or ITT was used, while in some instances they were rarely administered. The outcome of the model revealed extending baits to these areas and treating livestock with the same regimen as fly endemic areas could strategically improve the control. Here, we observed that *T. vivax* infection transmitted by both tsetse and biting flies would show $\mathscr{R}_0 > 1$, provided there are wildlife reservoir hosts present and abundant biting flies. It has long been reported that biting flies maintained *T. vivax* in cattle herd in areas with few or no tsetse flies [7]. During the hot dry season tsetse population reduces, however, there are abundant biting flies mechanically transmitting the disease, hence, *T. vivax* infection has been reported more often during the dry season [31]. Also, biting flies could develop resistance to commonly used insecticides in a geographical location based on quantity and frequency of use [33]. This resistance is less likely in tsetse flies because they are K-strategists, in which case they have low reproducing capacity with very high success rate [45].

We explained further in the model that if viable ITT are not in place, both in the targeted and adjacent areas, wildlife reservoirs could be a continuous source of infection to the livestock population from biting flies even when tsetse flies are absent. Furthermore, small ruminants in the SIR model revealed that in cases of sub-clinical infection, they could also act as reservoirs. The *T. brucei* situation in the model reflects the possibility of its elimination in cattle herds,

because its prevalence generally has been low. Besides, even when blood meals are exclusively from wildlife reservoirs by both tsetse and biting flies, the use of ITC and trypanocides on only the cattle population is effective and results in $R_{01} < 1$ for *T. b. brucei*. This report corroborated the result from a previous model [20]. This is probably due to management practices such as transhumance, in which the animals are constantly exposed to both infected tsetse flies and biting flies during migration.

However, the use of trypanocides and ITC cannot completely eliminate *T. congolense* and *T. vivax* (which seems to be the current control approach in West Africa). A low rate of infection could still persist, but the inclusion of ITT in the control strategy will make $R_{01} < 1$. This could be more effective if the insecticide choice is effective for both tsetse flies and biting flies. There could also be a need for extensive traps and baits for the vector flies. The presence of infected wildlife reservoirs could increase the basic reproduction number, except where cattle are permanently kept in an intensive system under optimal control conditions (trypanocides, traps, ITC, ITT and physical barriers).

There are high densities of cattle and transmitting-vectors of AAT in the project area from the field-data, allowing reliance on the use of insecticide-treated cattle (RAP method) as the simplest and cheapest option in parts of the operational area. However, the use of artificial baits and targets (ITT) is expected in the parts where cattle do not visit. To be most cost effective, each of the cattle due for insecticide and trypanocide treatment must be given a dose of a recommended insecticide some 4—12 times per year, depending on the level of infection. In this study, the cost of vector suppression / elimination was estimated at between 599—1875 US\$ / km$^2$, which was slightly higher than estimated cost of 200—1500 US\$ / km$^2$ in Uganda [46]. Also the control costs in southwest Nigeria were more than those estimated for southeastern Uganda [47]. This could be attributed to the number of livestock, management practices, manpower and density of flies. The insecticide used on cattle needs to be confirmed as acceptable to the national veterinary authorities to avoid insecticidal resistance and fly persistence, especially in biting flies [32]. It is expected that infected tsetse population would reduce due to the insecticide treatment, rather than the tsetse becoming infected through feeding. Hence, at the invasion front(s), where the cattle are liable to be challenged most often, the additional use of targets could deal with the greatest challenge. Besides, management practices such as pastoralism, nomadism and transhumance could have an effect in the model, and hence the frequency of ITC on fly control is essential in the control strategy.

Moreover, targets could be deployed in the invasion sources outside of the operational area, provided the baits are maintained at about 3 km$^2$ into the invasion area, then invasion stops completely. This integrated approach will contain all the transmitting vectors and protect the main target which is the cattle. The total cost of eliminating AAT in southwest Nigeria based on the model from this study is economical considering the impact on the livestock industry and could serve as a template for other parts of Africa, except for regions where *T. brucei rhodesiense* is prominent, a factor not included in this model.

Our model showed that apart from tsetse flies which are biological vectors, biting flies remain major drivers in maintaining *T. vivax* in West Africa due to their abundance, persistence, resilience to seasonal variations, resistance to insecticides and high reproductive capacity. The prevalence of AAT in vertebrate hosts and vectorial capacity of biological and mechanical vector flies were important factors in the elimination approach. Meanwhile, the southwest Nigeria field model validates our theoretical model because all the biological and mechanical vectors were observed to harbour trypanosomes, in which our model showed that they have the potential to transmit the pathogen. Eliminating tsetse flies which are K-strategists (species with low reproduction rate) and highly susceptible to insecticides [45, 48, 49], may not necessarily eliminate the biting flies. Hence, more studies are needed on biting flies that could

transmit trypanosomes from domestic vertebrates and reservoir hosts due to their interrupted feeding patterns, even when the biological vector is absent. Therefore, this model showed that there is a need to concentrate elimination programmes on all the fly vectors with transmission potentials, among other control plans, like the use of trypanocides and institutionalising barriers between domestic and sylvatic cycles.

## Conclusion

The strategic insecticidal approach (ITC and ITT as recommended) with periodic use of trypanocides, and further establishment of barriers between the domestic and sylvatic cycle, will improve cattle population and complete elimination of AAT. With the aid of suitable Lyapunov functions, the stability of the equilibria was explored. It was concluded from the analyses and simulations that if the intervention parameters $\mathscr{R}_0$, $R_{01}$ and $R_{02}$ are $<1$, the spread of African animal trypanosomosis decreases and could be sufficiently controlled.

## Appendix A: Proof of theorem 1

If the vertebrate hosts (cattle, small ruminants and wildlife) and invertebrate hosts (tsetse fly, stomoxyines, tabanids) population sizes are given by $N_k(t) = S_k(t) + E_k(t) + I_k(t) + R_k(t)$, $N_r(t) = S_r(t) + I_r(t) + R_r(t)$, $N_w(t) = S_w(t) + I_w(t)$ $N_t(t) = S_t(t) + E_t + I_t$, $N_s = S_n(t) + S_c$ and $N_b(t) = T_n(t) + T_c$. Then one see from (0.1)–(0.16)

$$\frac{dN_k(t)}{dt} \leq \Lambda_k - \mu_k N_k(t) \tag{0.21}$$

$$\frac{dN_t}{dt} \leq \Lambda_t - \mu_t N_t \tag{0.22}$$

$$\frac{dN_s}{dt} \leq \Lambda_s - \mu_s N_s \tag{0.23}$$

$$\frac{dN_b}{dt} \leq \Lambda_b - \mu_b N_s \tag{0.24}$$

$$\frac{dN_w}{dt} \leq \Lambda_w - \mu_w N_w \tag{0.25}$$

$$\frac{dN_r}{dt} \leq \Lambda_r - \mu_b N_r \tag{0.26}$$

solving the differential inequalities (0.21)–(0.25) and (0.26) one after the other gives

$$N_k(t)e^{\mu_k t} \leq N_k(0) + \frac{\Lambda_k}{\mu_k}e^{\mu_k t} - \frac{\Lambda_k}{\mu_k}$$

so that

$$N_k(t) \leq N_k(0)e^{-\mu_k t} + \frac{\Lambda_k}{\mu_k} - \frac{\Lambda_k}{\mu_k}e^{-\mu_k t}$$

this implies

$$N_k(t) \leq \frac{\Lambda_k}{\mu_k}(1 - e^{-\mu_k t}) + N_k(0)e^{-\mu_k t} \tag{0.27}$$

From (0.22)

$$N_t(t)e^{\mu_t t} \leq N_t(0) + \frac{\Lambda_t}{\mu_t}(e^{\mu_t t} - \frac{\Lambda_t}{(\mu_t}$$

so that

$$N_t(t) \leq N_t(0)e^{-\mu_t t} + \frac{\Lambda_t}{\mu_t} - \frac{\Lambda_t}{\mu_t}e^{-(\mu_t + \sigma_t)t}$$

this implies

$$N_t(t) \leq \frac{\Lambda_t}{\mu_t}(1 - e^{-(\mu_t t)}) + N_t(0)e^{-\mu_t t} \tag{0.28}$$

In a similar manner, Eqs (0.23)–(0.25) and (0.26) gives

$$N_s(t) \leq \frac{\Lambda_s}{\mu_s}(1 - e^{-\mu_s t}) + N_s(0)e^{-\mu_s t} \tag{0.29}$$

$$N_b(t) \leq \frac{\Lambda_b}{(\mu_b}(1 - e^{-(\mu_b + \delta_b)t}) + N_b(0)e^{-\mu_b t} \tag{0.30}$$

$$N_w(t) \leq \frac{\Lambda_w}{\mu_w}(1 - e^{-\mu_b t}) + N_w(0)e^{-\mu_w t} \tag{0.31}$$

$$N_r(t) \leq \frac{\Lambda_r}{\mu_r}(1 - e^{-\mu_r t}) + N_r(0)e^{-\mu_r t} \tag{0.32}$$

Taking the limits of (0.27)–(0.32) as $t \to \infty$ gives $N_k(t) \leq \frac{\Lambda_k}{\mu_k}$, $N_t(t) \leq \frac{\Lambda_t}{\mu_t}$, $N_s(t) \leq \frac{\Lambda_s}{\mu_s}$, $N_b(t) \leq \frac{\Lambda_b}{\mu_b}, \frac{\Lambda_w}{\mu_w}, \frac{\Lambda_r}{\mu_r}$. Thus the following feasible region

$$\mathscr{R} = \{S_k(t), E_k(t), I_k(t), R_k(t), S_t, E_t, I_t, S_n, S_c, T_n, T_c, S_w, I_w, S_r, I_r, R_r \in \mathscr{R}^{16} : N_k(t) \leq \frac{\Lambda_k}{\mu_k},$$

$$N_t(t) \leq \frac{\Lambda_t}{\mu_t}, N_s(t) \leq \frac{\Lambda_s}{\mu_s}, N_b(t) \leq \frac{\Lambda_b}{\mu_b}, N_w(t) \leq \frac{\Lambda_w}{\mu_w}, N_r(t) \leq \frac{\Lambda_r}{\mu_r}\}$$

## Appendix B: Proof of theorem 2

Consider the following linear Lyapunov function

$$F = \frac{\alpha_k E_k(t)}{\mu_t \mu_b \mu_s (\alpha_k + \mu_k)(\rho r_k + \tau_k + \mu_k)} + \frac{I_k(t)}{\mu_t \mu_b \mu_s (r_h(a_i) + \tau_k + \mu_k)} + \frac{E_t(t)}{b\tau\Lambda_t} + \frac{I_t(t)(\gamma_t + \sigma_t + \mu_t)}{b\gamma_t\Lambda_t}$$

$$+ \frac{S_c(t)(\mu_s + \delta_s)}{b\Lambda_s} + \frac{T_c(t)(\mu_b + \delta_b)}{b\Lambda_b} + \frac{I_w(t)}{\mu_w} + \frac{I_r(t)}{\theta + \phi + \mu_r} \tag{0.33}$$

In what follows, the time derivative of $F$ given by (0.33) along the solutions of the model (0.1)–(0.16) yields

$$
\begin{aligned}
\dot{F} = {} & \frac{\alpha_k}{\mu_t \mu_b \mu_s (\alpha_k + \mu_k)(\rho r_k + \tau_k + \mu_k)} + [b\varphi S_k(t)(I_t + T_c + S_c) - (\alpha_k + \mu_k)E_k(t)] + \frac{\alpha_k E_k(t) - (\rho r_k + \tau_k + \mu_k)I_k(t)}{\mu_t \mu_b \mu_s(\rho r_k + \tau_k + \mu_k)} \\
& + \frac{b\varphi S_k(t)(I_t + T_c + S_c) - (\gamma_t + \sigma_t + \mu_t)E_t(t)}{b\Lambda_t \tau} + \frac{(\sigma_t + \mu_t)[\gamma_t E_t(t) - (\sigma_t + \mu_t)I_t(t)]}{b\gamma_t \Lambda_t} \\
& + \frac{(\mu_s + \delta_s)[b\alpha_2 S_n(I_k + I_r + I_w) - (\mu_s + \delta_s)S_c]}{b\Lambda_s} + \frac{(\mu_b + \delta_b)[b\alpha_1 T_n(I_k + I_r + I_w) - (\mu_b + \delta_b)T_c]}{b\Lambda_b} \\
& + \frac{e^{\omega_2} b\zeta(I_t + T_c + S_c) - \mu_w}{\mu_w \Lambda_w} + \frac{e^{\omega_2 b}\tau(I_t + T_c + S_c) - (\theta + \phi + \mu_r)}{(\theta + \phi + \mu_r)}
\end{aligned}
\tag{0.34}
$$

Further simplification of $\dot{F}$ gives

$$
\begin{aligned}
\dot{F} \le {} & \frac{(\sigma_t + \mu_t)(\gamma_t + \sigma_t + \mu_t)(I_t + T_c + S_c)}{b\gamma_t \Lambda_t \Lambda_b \Lambda_s} \\
& \times \left[ e^{\omega_1 + \omega_2} \left( \frac{b^2 \alpha_k \varphi \tau \zeta \varphi_t \Lambda_k \Lambda_r \Lambda_w \Lambda_t \Lambda_s \Lambda_t \alpha_t \alpha_1 \alpha_2}{\mu_k \mu_t \mu_s \mu_b \mu_w \mu_r (\alpha_k + \mu_k)(\rho r_k + \tau_k + \mu_k)(\gamma_t + \sigma_t + \mu_t)(\sigma_t + \mu_t)(\mu_b + \delta_b)(\mu_s + \mu_s)(\theta + \phi + \mu - r)} \right) - 1 \right]
\end{aligned}
$$

$$
\dot{F} \le \frac{(\sigma_t + \mu_t)(\gamma_t + \sigma_t + \mu_t)}{b\gamma_t \Lambda_t \Lambda_b \Lambda_s}(R_{01} - 1)(I_t + T_c + S_c)
$$

$\dot{F} \le 0$ for $R_{01} \le 1$. $\dot{F} = 0$ if and only if $I_t(t) = T_c(t) = S_c(t) = 0$. Further, one sees that $(S_k(t), E_k(t), I_k(t), R_k(t), S_t(t), E_t(t), I_t(t), S_n(t), S_c, T_n, T_c, S_w, I_w, S_r, I_r, R_r) \to \pi_0 = (\frac{\Lambda_k}{\mu_k}, 0, 0, 0, \frac{\Lambda_t}{\mu_t}, 0, 0, \frac{\Lambda_s}{\mu_s}, 0, \frac{\Lambda_b}{\mu_b}, 0, \frac{\Lambda_w}{\mu_w}, 0, \frac{\Lambda_r}{\mu_r}, 0, 0)$ as $t \to \infty$ since $(I_t, T_c, S_c \to 0$ as $t \to \infty$. Consequently, the largest compact invariant set in $\{(S_k(t), E_k(t), I_k(t), R_k(t), S_t(t), E_t(t), I_t(t), S_n(t), S_c, T_n, T_c, S_w, I_w, S_r, I_r, R_r \in \mathscr{R} : \dot{F} = 0\}$ is a singleton $\{\pi_0\}$ and by LaSalle's invariance principle [43], $\pi_0$ is globally asymptotically stable in $\mathscr{R}$ if $R_{01} \le 1$.

## Appendix C: Proof of theorem 3

Let $\mathscr{R}_0 > 1$, $R_{01} > 1$ and $R_{02} > 1$ so that a unique endemic equilibrium exists and consider the following nonlinear Lyapunov function defined by

$$
\begin{aligned}
V = {} & S_k(t) - S_k^* - S_k^* \ln\left(\frac{S_h(t)}{S_k^*}\right) + E_k(t) - E_k^* - E_k^* \ln\left(\frac{E_k(t)}{E_k^{**}}\right) + \frac{(\rho r_k + \tau_k + \mu_k)}{\tau_k}\left[I_k(t) - I_k^* - I_k^* \ln\left(\frac{I_k(t)}{I_k^*}\right)\right] \\
& + S_t - S_t^* - S_t^{**} \ln\left(\frac{S_t}{S_m^*}\right) + E_t - E_t^* - E_t^* \ln\left(\frac{E_t}{E_t^*}\right) + \frac{(\sigma_t + \mu_t)}{\sigma_t}\left[I_t - I_t^* - I_t^* \ln\left(\frac{I_t}{I_t^*}\right)\right] \\
& + S_n - S_n^* - S_n^{**} \ln\left(\frac{S_n}{S_n^*}\right) + \frac{(\mu_s + \delta_s)}{\delta_s}\left[S_c - S_c^* - S_c^* \ln\left(\frac{S_c}{S_c^*}\right)\right] + T_n - T_n^* - T_n^{**} \ln\left(\frac{T_n}{T_n^*}\right) \\
& + \frac{(\mu_b + \delta_b)}{\delta_b}\left[T_c - T_c^* - T_c^* \ln\left(\frac{T_c}{T_c^*}\right)\right] + \frac{1}{\mu_w}\left[I_w(t) - I_w^* - I_w^* \ln\left(\frac{I_w(t)}{I_w^*}\right)\right] + \frac{\theta + \phi + \mu_r}{\phi}\left[I_r(t) - I_r^* - I_r^* \ln\left(\frac{I_r(t)}{I_r^*}\right)\right]
\end{aligned}
\tag{0.35}
$$

The Lyapunov derivative of (0.35) is given by

$$
\begin{aligned}
\dot{V} =\; & \Lambda_k\left(1 - \frac{S_k^*}{S_k(t)}\right) - \mu_k S_k(t)\left(1 - \frac{S_k^*}{S_k(t)}\right) + b\varphi S_k^*(I_t(t) + T_c(t) + S_c(t)) - \frac{b\varphi S_k(t)E_k^*(I_t(t) + T_c(t) + S_c(t))}{E_k(t)} \\
& + (\alpha_k + \mu_k)E_k^* + \frac{(\alpha_k + \mu_k)(\rho r_k + \tau_k + \mu_k)I_k(t)}{\tau_k} - \frac{(\alpha_k + \mu_k)I_k^* E_k(t)}{I_k(t)} + \frac{(\alpha_k + \mu_k)(\rho r_k + \tau_k + \mu_k)I_k^*}{\alpha_k} \\
& + \Lambda_t\left(1 - \frac{S_t^*}{S_t}\right) - \mu_t S_t\left(1 - \frac{S_t^*}{S_t}\right) + b\varphi_t S_t(t)(I_k(t) + I_w(t) + I_r(t)) - \frac{b\varphi_t S_t E_t^*(I_k(t) + I_w(t) + I_r(t))}{E_t} \\
& + (\gamma_t + \mu_t + \sigma_t)E_t^* - \frac{(\gamma_t + \mu_t + \sigma_t)(\sigma_t + \mu_t)I_t}{\gamma_t} - \frac{(\gamma_t + \mu_t + \sigma_t)I_t^* E_t}{I_t} + \frac{(\gamma_t + \mu_t + \sigma_t)(\sigma_t + \mu_t)I_t^*}{\alpha_t} \qquad (0.36) \\
& + \Lambda_w\left(1 - \frac{S_w^*}{S_w(t)}\right) - \mu_w\left(1 - \frac{S_w^*}{S_w(t)}\right) + b\tau e^{\omega_1}S_w(I_t + T_c + S_c) + \Lambda_r\left(1 - \frac{S_r^*}{S_r(t)}\right) - \mu_r\left(1 - \frac{S_r^*}{S_r(t)}\right) \\
& + b\zeta e^{\omega_2}S_r(I_t + T_c + S_c) + \Lambda_s\left(1 - \frac{S_n^*}{S_n(t)}\right) - \mu_s\left(1 - \frac{S_n^*}{S_n(t)}\right) + b\alpha_2 S_n(I_k + I_w + I_r) \\
& + \Lambda_b\left(1 - \frac{T_n^*}{T_n(t)}\right) - \mu_b\left(1 - \frac{T_n^*}{T_n(t)}\right) + b\alpha_1 T_n(I_k + I_w + I_r)
\end{aligned}
$$

At the endemic equilibrium, it is seen from (0.1)–(0.16) that

$$
\left.
\begin{aligned}
\Lambda_k &= b\varphi S_k^*(I_t^* + T_c^*(t) + S_c^*(t)) + \mu_k S_k^* \\[4pt]
\alpha_k + \mu_k &= \frac{b\varphi S_k^*(I_t^* + T_c^*(t) + S_c^*(t))}{E_k^*} \\[4pt]
\rho r_h + \tau_k + \mu_k &= \frac{\alpha_k E_k^*}{I_k^*} \\[4pt]
\Lambda_t &= b\varphi_t S_t^*(I_k^* + I_w^* + I_r^*) + \mu_t S_t^* \\[4pt]
\gamma_t + \sigma_t + \mu_t &= \frac{b\varphi_t S_t^*(I_k^* + I_w^* + I_r^*)}{E_t^*} \\[4pt]
\sigma_t + \mu_t &= \frac{\gamma E_t^*}{I_t^*} \\[4pt]
\Lambda_w &= e^{\omega_1}b\tau S_w(I_t^* + S_c^* + T_c^*) + \mu_w \\[4pt]
\Lambda_r &= e^{\omega_2}b\zeta S_r(I_t^* + S_c^* + T_c^*) + \mu_r \\[4pt]
\Lambda_s &= b\alpha_2 S_n(I_k^* + I_r^* + I_w^*) + (\mu_s + \delta_s)S_n^* \\[4pt]
\Lambda_s &= b\alpha_1 T_n(I_k^* + I_r^* + I_w^*) + (\mu_b + \delta_b)T_n^*
\end{aligned}
\right\}
\qquad (0.37)
$$

and using (0.37) in (0.36), and then add and subtract the following systematically

$b\varphi S_k^*(I_t^*) + T_c^*(t) + S_c^*(t))$, $b\varphi_t S_t^*(I_k^* + I_w^* + I_r^*)$, $\frac{b\varphi_k S_k^* I_k(t)(I_t^*) + T_c^*(t) + S_c^*(t))^2}{I_k^*(I_t + T_c(t) + S_c(t))}$, $\frac{b\varphi_t S_t^* I_t(I_k^* + I_w^* + I_r^*)^2}{I_t^{**}(I_k + I_w + I_r)}$,

$be^{\omega_1}\tau S_w(I_t^*) + T_c^*(t) + S_c^*(t))$, $be^{\omega_2}\zeta S_r(I_t^*) + T_c^*(t) + S_c^*(t))$, $\frac{be^{\omega_1}S_w I_w(I_t^*) + T_c^*(t) + S_c^*(t))^2}{I_w^*(I_t(t) + T_c(t) + S_c(t))}$,

$\frac{be^{\omega_2}S_r I_r(I_t^*) + T_c^*(t) + S_c^*(t))^2}{I_r^*(I_t(t) + T_c(t) + S_c(t))}$, $b\alpha_2 S_n(I_k^* + I_w^* + I_r^*)$, $b\alpha_1 T_n(I_k^* + I_w^* + I_r^*)$, $\frac{b\alpha_2 S_n S_c(I_k^* + I_w^* + I_r^*)^2}{S_c(I_k + I_w + I_r)}$, $\frac{b\alpha_1 T_n T_c(I_k^* + I_w^* + I_r^*)^2}{T_c(I_k + I_w + I_r)}$ one

gets

$$\dot{V} = \mu_k S_k^* \left( 2 - \frac{S_k^*}{S_k(t)} - \frac{S_k(t)}{S_k^*} \right) + b\varphi S_k^*(I_t^*) + T_c^*(t) + S_c^*(t)) \times$$

$$\left[ 4 - \frac{S_k^*)}{S_k(t)} - \frac{E_k^* S_k(t) I_t T_c S_c}{E_k(t) S_k^*) I_t^* T_c^* S_c^*} - \frac{I_k^* E_k(t)}{I_k(t) E_k^*} - \frac{I_k(t) I_t^* T_c^* S_c^*}{I_k^* I_t T_c S_c} \right]$$

$$+ b\varphi S_k^*(I_t + T_c + S_c) - \frac{b\varphi S_k^* I_k(t)(I_t^*) + T_c^*(t) + S_c^*(t))}{I_k^*} + \frac{b\varphi S_k^* I_k(t)(I_t^*) + T_c^*(t) + S_c^*(t))^2}{I_k^*(I_t) + T_c^*(t) + S_c^*(t))}$$

$$- b\varphi S_k^*(I_t^*) + T_c^*(t) + S_c^*(t)) + \mu_t S_t^* \left( 2 - \frac{S_t^*}{S_t} - \frac{S_t}{S_t^*} \right) + b\varphi_t S_t^*(I_k^* + I_w^* + I_r^*) \times$$

$$\left[ 4 - \frac{S_t^*}{S_t} - \frac{E_t^* S_t(I_k + I_w + I_r)}{E_t S_t^*(I_k^* + I_w^* + I_r^*)} - \frac{I_t^* E_t}{I_t E_t^*} - \frac{(I_k^* + I_w^* + I_r^*) I_t}{I_t^*(I_k + I_w + I_r)(t)} \right]$$

$$+ b\varphi_t S_t^*(I_k + I_w + I_r) - \frac{b\varphi_t S_t^*(I_k^* + I_w^* + I_r^*) I_t}{I_t^*} + \frac{b\varphi_t S_t^* I_t(I_k^* + I_w^* + I_r^*)^2}{I_t^*} - b\varphi_t S_t^*(I_k^* + I_w^* + I_r^*)$$

$$+ \mu_w S_w^* \left( 2 - \frac{S_w^*}{S_w(t)} - \frac{S_w(t)}{S_w^*} \right) + be^{\omega_1}\tau S_w^*(I_t^*) + T_c^*(t) + S_c^*(t)) \times$$

$$\left[ 4 - \frac{S_w^*)}{S_w(t)} - \frac{S_w(t) I_t T_c S_c}{S_w^*) I_t^* T_c^* S_c^*} - \frac{I_w^*}{I_w(t)} - \frac{I_w(t) I_t^* T_c^* S_c^*}{I_w^* I_t T_c S_c} \right] + \mu_r S_r^* \left( 2 - \frac{S_r^*}{S_r(t)} - \frac{S_r(t)}{S_r^*} \right) + be^{\omega_2}\zeta S_r^*(I_t^*) + T_c^*(t) + S_c^*(t)) \times$$

$$\left[ 4 - \frac{S_r^*)}{S_r(t)} - \frac{S_r(t) I_t T_c S_c}{S_w^*) I_t^* T_c^* S_c^*} - \frac{I_r^*}{I_r(t)} - \frac{I_r(t) I_t^* T_c^* S_c^*}{I_r^* I_t T_c S_c} \right] + \mu_s S_n^* \left( 2 - \frac{S_n^*}{S_n} - \frac{S_n}{S_n^*} \right) + b\alpha_2 S_t^*(I_k^* + I_w^* + I_r^*) \times$$

$$\left[ 4 - \frac{S_n^*}{S_n} - \frac{S_n(I_k + I_w + I_r)}{S_n^*(I_k^* + I_w^* + I_r^*)} - \frac{S_c^*}{S_c} - \frac{(I_k^* + I_w^* + I_r^*) S_c}{S_c^*(I_k + I_w + I_r)(t)} \right] + \mu_b T_n^* \left( 2 - \frac{T_n^*}{T_n} - \frac{T_n}{T_n^*} \right) + b\alpha_1 T_n^*(I_k^* + I_w^* + I_r^*) \times$$

$$\left[ 4 - \frac{T_n^*}{T_n} - \frac{T_n(I_k + I_w + I_r)}{T_n^*(I_k^* + I_w^* + I_r^*)} - \frac{T_c^*}{T_c} - \frac{(I_k^* + I_w^* + I_r^*) T_c}{T_c^*(I_k + I_w + I_r)(t)} \right]$$

simplify further, we have

$$\dot{V} = -V_1 - V_2 - b\varphi S_k^*(I_t^* + T_c^* + S_c^*) \left[ 1 - \frac{I_t T_c S_c}{I_t^* T_c^* S_c^*} + \frac{I_k(t)}{I_k^*} - \frac{I_k(t) I_t^* T_c^* S_c^*}{I_k^* I_t T_c S_c} \right]$$

$$-V_3 - V_4 - e^{\omega_1} b\tau S_w(I_t + T_c + S_c) \left[ 1 - \frac{I_t T_c S_c}{I_t^* T_c^* S_c^*} + \frac{I_w(t)}{I_w^*} - \frac{I_w(t) I_t^* T_c^* S_c^*}{I_w^* I_t T_c S_c} \right]$$

$$-v_5 - V_6 - e^{\omega_2} b\zeta S_r(I_t + T_c + S_c) \left[ 1 - \frac{I_t T_c S_c}{I_t^* T_c^* S_c^*} + \frac{I_r(t)}{I_r^*} - \frac{I_r(t) I_t^* T_c^* S_c^*}{I_r^* I_t T_c S_c} \right] \qquad (0.38)$$

$$-V_7 - V_8 - b\varphi_t S_t(I_k + I_w + I_r) \left[ 1 - \frac{I_k I_w I_r}{I_k^* I_w^* I_r^*} + \frac{I_t T_c S_c}{I_t^* T_c^* S_c^*} - \frac{I_t T_c S_c I_k^* I_w^* I_r^*}{I_t^* T_c^* S_c^* I_k I_w I_r} \right]$$

$$-V_9 - V_{10} - b\alpha_1 T_n(I_k + I_w + I_r) \left[ 1 - \frac{I_k I_w I_r}{I_k^* I_w^* I_r^*} + \frac{I_t T_c S_c}{I_t^* T_c^* S_c^*} - \frac{I_t T_c S_c I_k^* I_w^* I_r^*}{I_t^* T_c^* S_c^* I_k I_w I_r} \right]$$

$$-V_{11} - V12 - b\alpha_2 S_n(I_k + I_w + I_r) \left[ 1 - \frac{I_k I_w I_r}{I_k^* I_w^* I_r^*} + \frac{I_t T_c S_c}{I_t^* T_c^* S_c^*} - \frac{I_t T_c S_c I_k^* I_w^* I_r^*}{I_t^* T_c^* S_c^* I_k I_w I_r} \right]$$

where $V_1 = \left( \frac{S_k^*}{S_k(t)} + \frac{S_k(t)}{S_k^*} - 2 \right)$, $V_2 = b\varphi S_k^*(I_t^*) + T_c^*(t) + S_c^*(t)) \times$
$\left[ \frac{S_k^*)}{S_k(t)} + \frac{E_k^* S_k(t) I_t T_c S_c}{E_k(t) S_k^*) I_t^* T_c^* S_c^*} + \frac{I_k^* E_k(t)}{I_k(t) E_k^*} + \frac{I_k(t) I_t^* T_c^* S_c^*}{I_k^* I_t T_c S_c} - 4 \right]$ $V_3 = \left( \frac{S_t^*}{S_t} + \frac{S_t}{S_t^*} - 2 \right)$,
$V_4 = b\varphi_t S_t^*(I_k^* + I_w^* + I_r^*) \left[ \frac{S_t^*}{S_t} + \frac{E_t^* S_t(I_k + I_w + I_r)}{E_t S_t^*(I_k^* + I_w^* + I_r^*)} + \frac{I_t^* E_t}{I_t E_t^*} + \frac{(I_k^* + I_w^* + I_r^*) I_t}{I_t^*(I_k + I_w + I_r)(t)} - 4 \right]$, $V_5 = \left( \frac{S_w^*}{S_w(t)} + \frac{S_w(t)}{S_w^*} - 2 \right)$,

$$V_6 = be^{\omega_1}\tau S_w^*(I_t^*) + T_c^*(t) + S_c^*(t))\left[\frac{S_w^*)}{S_w(t)} + \frac{S_w(t)I_t T_c S_c}{S_w^*)I_t^* T_c^* S_c^*} + \frac{I_w^*}{I_w(t)} + \frac{I_w(t)I_t^* T_c^* S_c^*}{I_w^* I_t T_c S_c} - 4\right],$$

$$V_7 = \left(\frac{S_r^*}{S_r(t)} + \frac{S_r(t)}{S_r^*} - 2\right), V_8 = be^{\omega_2}\zeta S_r^*(I_t^*) + T_c^*(t) + S_c^*(t))\left[\frac{S_r^*)}{S_r(t)} + \frac{S_r(t)I_t T_c S_c}{S_w^*)I_t^* T_c^* S_c^*} + \frac{I_r^*}{I_r(t)} + \frac{I_r(t)I_t^* T_c^* S_c^*}{I_r^* I_t T_c S_c} - 4\right],$$

$$v_9 = \left(2 - \frac{S_n^*}{S_n} - \frac{S_n}{S_n^*}\right), V_{10} = b\alpha_2 S_t^*(I_k^* + I_w^* + I_r^*)\left[\frac{S_n^*}{S_n} + \frac{S_n(I_k + I_w + I_r)}{S_n^*(I_k^* + I_w^* + I_r^*)} + \frac{S_c^*}{S_c} + \frac{(I_k^* + I_w^* + I_r^*)S_c}{S_c^*(I_k + I_w + I_r)(t)} - 4\right],$$

$$V_{11} = \left(2 - \frac{T_n^*}{T_n} - \frac{T_n}{T_n^*}\right), V_{12} = b\alpha_1 T_n^*(I_k^* + I_w^* + I_r^*)\left[\frac{T_n^*}{T_n} + \frac{T_n(I_k + I_w + I_r)}{T_n^*(I_k^* + I_w^* + I_r^*)} + \frac{T_c^*}{T_c} + \frac{(I_k^* + I_w^* + I_r^*)T_c}{T_c^*(I_k + I_w + I_r)(t)} - 4\right]$$

We need to show that $V_i \geq 0$, Â $i = 1, 2, \ldots 12$,. In order to achieve this model, considering the arithmetic mean is greater than or equal to the geometric mean (AM—GM inequality), we have $(S_k^*)^2 + (S_k(t))^2 - 2S_k^*S_k(t) \geq 0$ so that, $\left(\frac{S_k^*}{S_k(t)} + \frac{S_k(t)}{S_k^*} - 2\right) \geq 0$. Hence, $V_1 \geq 0$. Further, let $x = \frac{S_k^*}{S_k(t)}, y = \frac{E_k^* I_t}{E_k(t)I_t^*}, z = \frac{I_k^* I_t}{I_k(t)I_t^*}$. Then, $b\varphi S_k^*(I_t^*) + T_c^*(t) + S_c^*(t)) \times$

$\left[\frac{S_k^*)}{S_k(t)} + \frac{E_k^* S_k(t)I_t T_c S_c}{E_k(t)S_k^*)I_t^* T_c^* S_c^*} + \frac{I_k^* E_k(t)}{I_k(t)E_k^*} + \frac{I_k(t)I_t^* T_c^* S_c^*}{I_k^* I_t T_c S_c} - 4\right]$ can be written as further algebraic manipulation gives

$$f(x, y, z) = x + \frac{y}{x} + \frac{z}{y} + \frac{1}{z} - 4 \tag{0.39}$$

It is suffice to show that $f(x, y, z) \geq 0$. Since $f_x = f_y = f_z = 0$ gives rise to $x = y = z$ and that $f_{xx} > 0, f_{yy} > 0, f_{zz} > 0$, one see that the minimum of $f(x, y, z)$ is attainable at $x = y = z$. In what follows, (0.39) is reduced to $(x-1)^2 \geq 0$ or $(y-1)^2 \geq 0$ or $(z-1)^2 \geq 0$ with equality if and only if $x = 1$ or $y = 1$ or $z = 1$ respectively. Hence, $V_2 \geq 0$. The proof of $V_3 \geq 0$ is similar to $V_1 \geq 0$ while that of $V_4 \geq 0$ is similar to $V_2 \geq 0$ and so on. Whenever $\frac{I_k^*}{I_k(t)} = \frac{I_t^*}{I_t} = \frac{T_c^*}{T_c} = \frac{S_c^*}{S_c}, \frac{I_w^*}{I_w(t)} = \frac{I_t^*}{I_t} = \frac{T_c^*}{T_c} = \frac{S_c^*}{S_c}, \frac{I_r^*}{I_r(t)} = \frac{I_t^*}{I_t} = \frac{T_c^*}{T_c} = \frac{S_c^*}{S_c}$ it follows from (0.38) that $\dot{V} \leq 0$ with $\dot{V} = 0$ if and only if $S_k(t) = S_k^*(t), E_k(t) = E_k^*, I_k(t) = I_k^*, S_t = S_t^*, E_t = E_t^*, I_t = I_t^*, S_n = S_n^*, S_c = S_c^*, T_n = T_n^*, T_c = T_c^* S_w = S_w^*, S_r = s_r^*, I_w = I_w^*, I_r = I_r^*$. This further implies that $R_k(t) = \frac{\rho r_k I_k^*}{\mu_k} = R_k^*, \frac{\theta R_r^*}{\mu_r} = R_r^*$ since $(S_k(t), E_k(t), I_k(t), S_w, I_w, S_r, I_r)$ tends to $(S_t^*, E_t^*, I_t^*, S_n^*, S_c^*), T_n^*, T_c^*$ as $t \to \infty$. Therefore, LaSalle's principle explains that the largest compact invariant subset of the set where $\dot{V} = 0$ is the endemic equilibrium point $E_e^*$. Hence, every solution in $\mathscr{R}$ approaches $E_e^*$ for $R_{01}, R_{02} > 1$, and $E_e^*$ is globally asymptotically stable.

## Supporting information

**S1 Fig. Vegetation of operational and adjacent areas to be baited.**
(PDF)

**S2 Fig. Existing problems or adversities affecting vector flies in operational and adjacent areas to be baited.**
(PDF)

**S3 Fig. Pretreatment population in operational and adjacent areas to be baited.**
(PDF)

**S4 Fig. Total areas estimated to be baited.**
(PDF)

**S1 Table. Elimination cost of tsetse vector in southwest Nigeria.**
(DOCX)

**S2 Table. Elimination strategies for tsetse flies in cattle rearing areas of southwest Nigeria using the tsetse model approach.**
(DOCX)

## Acknowledgments

We appreciate Mr. Deji for designing the Nzi traps, Mr. Amidu for access to Fulani farm settlements and Mr Suleiman for providing cattle for insecticidal experiment. Gratitude to Ms Kehinde Omolabi for proof-reading this article. We appreciate Professor Glyn Vale and team members for designing Tsetse Plan, and making the software available.

## Author Contributions

**Conceptualization:** Paul Olalekan Odeniran.

**Data curation:** Paul Olalekan Odeniran, Akindele Akano Onifade.

**Formal analysis:** Akindele Akano Onifade, Simon Alderton.

**Funding acquisition:** Susan Christina Welburn.

**Investigation:** Isaiah Oluwafemi Ademola.

**Methodology:** Akindele Akano Onifade.

**Project administration:** Paul Olalekan Odeniran.

**Resources:** Akindele Akano Onifade, Ewan Thomas MacLeod, Susan Christina Welburn.

**Software:** Akindele Akano Onifade, Ewan Thomas MacLeod, Susan Christina Welburn.

**Supervision:** Ewan Thomas MacLeod, Isaiah Oluwafemi Ademola, Susan Christina Welburn.

**Validation:** Ewan Thomas MacLeod, Isaiah Oluwafemi Ademola, Simon Alderton, Susan Christina Welburn.

**Visualization:** Ewan Thomas MacLeod, Isaiah Oluwafemi Ademola, Simon Alderton, Susan Christina Welburn.

**Writing – original draft:** Paul Olalekan Odeniran, Akindele Akano Onifade.

**Writing – review & editing:** Paul Olalekan Odeniran, Akindele Akano Onifade, Ewan Thomas MacLeod, Isaiah Oluwafemi Ademola, Simon Alderton, Susan Christina Welburn.

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
