## [Decision Letter · Decision Letter 0]

22 Jul 2020

PONE-D-20-18752

Mathematical modelling and control of African animal trypanosomosis with interacting populations in West Africa- could biting flies be important in maintaining the disease endemicity?

PLOS ONE

Dear Dr. Odeniran,

Thank you for submitting your manuscript to PLOS ONE. After careful consideration, we feel that it has merit but does not fully meet PLOS ONE’s publication criteria as it currently stands. Therefore, we invite you to submit a revised version of the manuscript that addresses the points raised during the review process.

Many thanks for submitting your manuscript to PLOS One

It was reviewed by two experts in the field who have suggested some revisions be made prior to acceptance.

The reviewers also suggest a copy editing review for English language

If you could write a response to reviewers that will help to expedite revision upon resubmission

I wish you the best of luck with your revisions

Hope you are keeping safe and well in these difficult times

Thanks

Simon

We look forward to receiving your revised manuscript.

Kind regards,

Simon Clegg, PhD

Academic Editor

PLOS ONE

Reviewers' comments:

Reviewer's Responses to Questions

**Comments to the Author**

1. Is the manuscript technically sound, and do the data support the conclusions?

Reviewer #1: Partly

Reviewer #2: Partly

2. Has the statistical analysis been performed appropriately and rigorously? 

Reviewer #1: N/A

Reviewer #2: I Don't Know

3. Have the authors made all data underlying the findings in their manuscript fully available?

Reviewer #1: Yes

Reviewer #2: Yes

4. Is the manuscript presented in an intelligible fashion and written in standard English?

Reviewer #1: No

Reviewer #2: No

5. Review Comments to the Author

Reviewer #1: 1. The manuscript should be revisited to improve language use. There are a number of confusing non-SVO formated sentences. Just to mention a few, see lines 26-28,39-41, 60-61, 73-76, 79-81, 104-106, 133-153, 180-194, 199-201, 207-210,222-231, 246-247 etc. There is also a great deal of terminology misuse! Just to mention a few; transmitting vectors [may be this means AAT biological and mechanical vectors]. As such some of the section headings e.g. lines 172-173 are equally not understandable

2. Abstract contains abbreviations that are not written in full first; ITT/ITC

3. The manuscript is incoherent with respect to the mathematical modeling, tsetse plan control and cost simulations and the prevalence studies. It is very hard to rationalize why this paper contains these three sections; these datasets and their implications on AAT transmission dynamics and control are not tied together.

4. Unless it is a peculiar AAT eco-epidemiological situation in Nigeria for which you need to provide references, sheep and goats suffer from clinical AAT. It is therefore surprising that they are classified together with wild animals and treated as mere reservoirs of AAT in this manuscript. Small ruminants should therefore be modeled as part of the AAT domestic cycle.

5. The expression for R_0 given on page 6 seems not to be coming from the proposed model. A citation for this expression should be provided. There is also no link between this expression and the proposed model.

6. Some equations on page 7 like (0.4) and (0.5) are missing the derivative sign

7. The model development indicates that tsetse flies get infected when feeding on infected cattle, wildlife and small ruminants (page 6, line 126). This is not reflected in the model. What I can see in the model is that tsetse flies get infection from infected cattle only. Basically the model needs to be revised. Adding a model diagram will also be helpful.

8. There is no derivation given for expressions of the basic reproduction numbers R_01 and R_02. On page 12 after equation (0.12), the authors indicate that the next generation matrix technique earlier described was used. The earlier description of this technique cannot be seen. This article can be used for this technique. Please refer to; *Van den Driessche, P., & Watmough, J. (2002). Reproduction numbers and sub-threshold endemic equilibria for compartmental models of disease transmission. Math. Biosci., 180, 29–48.

9. One page 5, line 114, the cattle variables are defined, the same should be done for the variables of other populations considered in the model for easy follow up. All model variables need to be defined.

10. Results for local stability of the disease-free equilibrium and endemic equilibrium are not shown. The following articles can assist in doing both local and global stability analysis of the disease-free and endemic equilibrium. Please refer to;

a) *Chavez, C. C., Feng, Z., & Huang, W. (2002). On the computation of R0 and its role on global stability. Mathematical Approaches for Emerging and Re-Emerging Infection Diseases: An Introduction. The IMA Volumes in Mathematics and Its Applications, 125, 31-65.

b) *McCluskey C. C.(2006), Lyapunov functions for tuberculosis models with fast and slow progression. Mathematical Biosciences and Engineering, 3(4), 603C614

c) *Korobeinikov A. (2004), Lyapunov functions and global properties for SEIR and SEIS epidemic models. Mathematical Medicine & Biology: A Journal of the IMA. 21(2)(2004).

d) Other similar publications on local and global stability analysis can be downloaded online.

11. For AAT, recovered animals become susceptible again. Additionally, an animal infected with T. b. brucei, for example, is susceptible to other trypanosome species. This has not been taken care of in the mathematical model

12. There is apparent lack of methods description with regard to both prevalence and tsetse plan [control options and their relative costs].

13. Presentation of results should be improved by only presenting the model outputs. All models and their parameterization should be presented in methods section. As well, the last section of results; lines 298-312 reads like methods and not results.

14. Insecticidal resistance is not known for tsetse; Tsetse are R-strategists [first paragraph of results section

15. Please quantify the effectiveness of ITC/ITT eluded in lines 239-240 and elsewhere in this manuscript

16. There are generic statements e.g. “ behavior of tsetse”, behavior of cattle, “the magnitude of the exposed cattle ….in lines 274-293 that render almost all that section impossible to comprehend

17. This MS treats tsetse and biting flies as equal vectors [mechanical or biological] without due regard to the type of AAT. T. brucei and T. congolense are biologically transmitted by tsetse and T. vivax by both mechanical and biological transmission! I don’t see this taken care of at model parameter description, setup and parameterization. As a result, numerical results presented in lines 274-293 don’t seem to make lots of epidemiological sense to me. Please also check that discussion lines 322-331, 340-359 make epidemiological sense. There are lots of inaccuracies presented in these lines.

18. Table 2; check that “whole fly” and “troublesome” make sense in the context of this MS

19. For Tables S1 and S2: Once you have provided a detailed methods section for these outputs, the results presented in these tables should give the reader an indication of the cost per animal per year [parasite control costs] or cost for vector control /km2 for a specified time needed for suppression or elimination. These costs should be discussed citing an existing body of knowledge about costs. The reader’s attention should be drawn to the methods of cost estimation deployed in tsetse plan to those implemented elsewhere before such costs can be compared.

Reviewer #2: I found it difficult to engage with the study, and I believe this is down in part to the structure of the paper and some missing information.

I will give a small number of examples.

In the Materials and Methods, the section on the Nigeria case study (starting on page 8) contains a lot of information which discusses previous work, without being clear what the study you are reporting on does in this area (either materials or methods). What is the tsetse software? and tsetse plan? How do they work? Are they the implementations of your model? Or are values derived from the model being fed into them to generate outputs? Does using this software validate the model in some way? Or extend it's outputs (eg into the economics or operational planning).

I have spent a lot of time going backwards and forwards trying to find bits of information, or where things were first described. Possibly my own fault for printing the paper rather than viewing on screen. However, “To obtain R0 for the the model (2.1)-(2.11), the next generation matrix technique earlier described were utilised” appear on page 12 but the only previous reference to next generation matrix technique is in the abstract. So if it is described it isn't clearly labelled as such.

The model equations are referenced in numerous places as (2.1)-(2.11), but in my copy of the manuscript the equations appear to be labelled (0.1)-(0.11).

There are a large number of grammatical and spelling mistakes, which I am unwilling to spend time on now given that the paper requires a substantial restructuring to separate material into its appropriate section and clarify how the different aspects of the study tie together. On next review, I would spend more time on the language.

6. PLOS authors have the option to publish the peer review history of their article (what does this mean?). If published, this will include your full peer review and any attached files.

Reviewer #2: No

---

## [Author Response · Author response to Decision Letter 0]

11 Aug 2020

Dear Editor,

Please we have attended to each of the responses from the reviewers below. We hope that this would be satisfactory.

Many thanks

Paul (Corresponding author). 

Reviewer #1: 

1. The manuscript should be revisited to improve language use. There are a number of confusing non-SVO formated sentences. Just to mention a few, see lines 26-28,39-41, 60-61, 73-76, 79-81, 104-106, 133-153, 180-194, 199-201, 207-210,222-231, 246-247 etc. There is also a great deal of terminology misuse! Just to mention a few; transmitting vectors [may be this means AAT biological and mechanical vectors]. As such some of the section headings e.g. lines 172-173 are equally not understandable

Response: We have corrected the grammatical languages as mentioned by the reviewers.

2. Abstract contains abbreviations that are not written in full first; ITT/ITC

Response: ITT/ITC- means insecticidal treated targets and insecticidal treated cattle. This has been included in the text.

3. The manuscript is incoherent with respect to the mathematical modeling, tsetse plan control and cost simulations and the prevalence studies. It is very hard to rationalize why this paper contains these three sections; these datasets and their implications on AAT transmission dynamics and control are not tied together.

Response: We have improved on the manuscript and also include an aspect in the discussion that reads “Our model showed that apart from tsetse flies which are biological vectors, biting flies remain major drivers in maintaining AAT in West Africa due to their abundance, persistence, resilience to seasonal variations, resistance to insecticides and high reproductive capacity. The prevalence of AAT in vertebrate host and vectoral capacity of biological and mechanical vector flies were important factors in elimination approach. Meanwhile, the southwest Nigeria field model validates our theoretical model because the all the biological and mechanical vectors were observed to harbour trypanosomes, in which our model showed that they have the potential to transmit the pathogen. Hence, eliminating tsetse flies which are K-strategist (species with low reproduction rate) and highly susceptible to insecticides [43, 44], may not necessarily eliminate the biting flies. Hence, more studies are needed on biting flies that could transmit trypanosomes from domestic vertebrates and reservoir hosts due to their interrupted feeding patterns, even when biological vector is absent. Therefore, this model showed that there is a need to concentrate elimination programmes on all the fly vectors with transmission potentials among other control plans like use of trypanocides and institutionalising barriers between domestic and sylvatic cycles”.

4. Unless it is a peculiar AAT eco-epidemiological situation in Nigeria for which you need to provide references, sheep and goats suffer from clinical AAT. It is therefore surprising that they are classified together with wild animals and treated as mere reservoirs of AAT in this manuscript. Small ruminants should therefore be modeled as part of the AAT domestic cycle.

Response: Small ruminants develop clinical AAT of heavy challenge of trypanosomes. In endemic region, small ruminants need to be treated to be in recovered class. Although, in the case of persistent, those in recovered class could return to susceptible class if exposed to trypanosome infection. Meanwhile, at subclinical level of trypanosome infection, they are thought to maintain the disease in the domestic cycle, serving as reservoirs for Trypanosoma species. These peculiar AAT eco-epidemiological situation in West Africa could largely be attributed to breed selection over the years. They rightly fit in SIR model, which has now been included in the work.

5. The expression for R_0 given on page 6 seems not to be coming from the proposed model. A citation for this expression should be provided. There is also no link between this expression and the proposed model.

Response: We have decided to develop an expression for R_0 in the model now.

6. Some equations on page 7 like (0.4) and (0.5) are missing the derivative sign

Response: The derivative sign has now been included as 〖dR〗_k/dt and 〖dS〗_t/dt in equation 0.4 and 0.5

7. The model development indicates that tsetse flies get infected when feeding on infected cattle, wildlife and small ruminants (page 6, line 126). This is not reflected in the model. What I can see in the model is that tsetse flies get infection from infected cattle only. Basically, the model needs to be revised. Adding a model diagram will also be helpful.

Response: Thank you for the observation. Additional models including infection of tsetse flies and biting flies from infected small ruminants and wildlife has been included (Equation 0.12 – 0.16). Moreover, we have included a schematic diagram for easy explanation of the model).

8. There is no derivation given for expressions of the basic reproduction numbers R_01 and R_02. On page 12 after equation (0.12), the authors indicate that the next generation matrix technique earlier described was used. The earlier description of this technique cannot be seen. This article can be used for this technique. Please refer to; *Van den Driessche, P., & Watmough, J. (2002). Reproduction numbers and sub-threshold endemic equilibria for compartmental models of disease transmission. Math. Biosci., 180, 29–48.

Response: We have shown the derivation and included the techniques of Diekmann et al., 1990 and Van den Driessche, P., and Watmough, J. (2002). These two authors have been cited, and the technique was applied to our model. 

9. One page 5, line 114, the cattle variables are defined, the same should be done for the variables of other populations considered in the model for easy follow up. All model variables need to be defined.

Response: We have defined other variables e.g. small ruminants and wildlife properly.

10. Results for local stability of the disease-free equilibrium and endemic equilibrium are not shown. The following articles can assist in doing both local and global stability analysis of the disease-free and endemic equilibrium. Please refer to;

a) *Chavez, C. C., Feng, Z., & Huang, W. (2002). On the computation of R0 and its role on global stability. Mathematical Approaches for Emerging and Re-Emerging Infection Diseases: An Introduction. The IMA Volumes in Mathematics and Its Applications, 125, 31-65.

b) *McCluskey C. C.(2006), Lyapunov functions for tuberculosis models with fast and slow progression. Mathematical Biosciences and Engineering, 3(4), 603C614

c) *Korobeinikov A. (2004), Lyapunov functions and global properties for SEIR and SEIS epidemic models. Mathematical Medicine & Biology: A Journal of the IMA. 21(2)(2004).

d) Other similar publications on local and global stability analysis can be downloaded online.

Local stability of an equilibrium would imply existence of that situation only for a short time (depending on certain circumstances or conditions). Whereas, global stability would imply existence of a situation regardless of any condition. For example, 1. If an endemic equilibrium is globally stable, it would imply that in the long run, the disease prevails (i.e. it is not cured- it becomes endemic- it is not eliminated) 2. If a disease-free equilibrium is globally stable it would imply that the disease finally dies out. 3. If a disease-free equilibrium is locally stable, it would imply that the disease would be eliminated (provided that certain conditions are met or for a short time depending on certain conditions). Mathematically, when global stability is performed, there is no need to perform local stability. However, in our-build-up on the global stability analyses of disease-free and endemic equilibrium, we have stated the local stability of disease-free and endemic equilibrium theorem, which implies the stability. We have also cited the suggested authors.

11. For AAT, recovered animals become susceptible again. Additionally, an animal infected with T. b. brucei, for example, is susceptible to other trypanosome species. This has not been taken care of in the mathematical model

Response: Thank you for your response. The model illustrates how the recovered class could become susceptible again. We have indicated it properly in the diagram (Fig 1). If an animal is infected with T. b. brucei and also susceptible to other trypanosome species, the recovery route is very essential. For instance, if infected animal is treated with trypanocide, the drug protects it from getting infected to other species within the protective period. We have explained this properly in the work.

12. There is apparent lack of methods description with regard to both prevalence and tsetse plan [control options and their relative costs].

Response: The methods on prevalence studies and control plans (which incorporated relative costs) have been properly explained.

13. Presentation of results should be improved by only presenting the model outputs. All models and their parameterization should be presented in methods section. As well, the last section of results; lines 298-312 reads like methods and not results.

Response: All models and parameterization have now been moved to the method section. The last section of the results on budgeting and compiling final report has been improved upon and now reads like results.

14. Insecticidal resistance is not known for tsetse; Tsetse are R-strategists [first paragraph of results section.

Response: The insecticide resistance was meant for biting flies which could pose continuous challenge of trypanosome infections. Tsetse are K-strategists, which have been properly explained in the last paragraph of the discussion.

15. Please quantify the effectiveness of ITC/ITT eluded in lines 239-240 and elsewhere in this manuscript.

Response: We have explained the differences in effectiveness of ITC/ITT in the manuscript. It has been recast thus “Due to pastoralism and nomadism management system widely practised in West Africa, ITC is very effective because livestock farmers move their animals around for grazing. For ITT, beyond the target animals were considered, in which the vector fly populations of adjacent areas were also targeted. This is an effective method in elimination strategy, it also targets flies within and without target areas. It is most effective in areas where zero-grazing is practised. The use of ITC was more effective than ITT which affected the feeding tsetse and other biting flies, largely because of management practices”.

16. There are generic statements e.g. “behavior of tsetse”, behavior of cattle, “the magnitude of the exposed cattle ….in lines 274-293 that render almost all that section impossible to comprehend.

Response: The word behaviour is attributed to the graph-based representation, while magnitude expresses the size number of the object in mathematics. We have explained this at the first mention for the readers.

17. This MS treats tsetse and biting flies as equal vectors [mechanical or biological] without due regard to the type of AAT. T. brucei and T. congolense are biologically transmitted by tsetse and T. vivax by both mechanical and biological transmission! I don’t see this taken care of at model parameter description, setup and parameterization. As a result, numerical results presented in lines 274-293 don’t seem to make lots of epidemiological sense to me. Please also check that discussion lines 322-331, 340-359 make epidemiological sense. There are lots of inaccuracies presented in these lines.

Response: We have improved the discussion. We earlier stated that contaminated biting flies only transmit T. vivax.

18. Table 2; check that “whole fly” and “troublesome” make sense in the context of this MS

Response: We have removed the whole fly and corrected troublesome to trypanosome.

19. For Tables S1 and S2: Once you have provided a detailed methods section for these outputs, the results presented in these tables should give the reader an indication of the cost per animal per year [parasite control costs] or cost for vector control /km2 for a specified time needed for suppression or elimination. These costs should be discussed citing an existing body of knowledge about costs. The reader’s attention should be drawn to the methods of cost estimation deployed in tsetse plan to those implemented elsewhere before such costs can be compared.

Response: This has been included in the method section, “The cost analyses is technique dependent. In this study two viable techniques in West Africa considered in the model were trapping, ITC and ITT. In cases of animal movement where sufficient livestock are present for ITC, insecticides are often applied by spraying (RAP has been cost effective) or pour on. The cost of traps is relatively high (not necessarily the cost of acquiring the traps), because of the cost of manpower required for deployment which is dependent on density. Notably, the targeted vector flies were Glossina species and biting flies, which means that combination of techniques were required”. 

The result section has also been updated with “We observed the cost for elimination of an isolated study area with Palpalis group of tsetse flies and biting flies to cost between 599 - 1875 US$ / km2. However, to maintain barriers against reinvasion for the next five years, the cost could add between 20 - 50% to the original cost. Meanwhile, if barriers are extended to larger areas for longer period, cost would increase further. The treatment of cattle in protected areas would be between 4 - 12 times in fly belt zones, depending on the severity of fly challenge from Glossina species and biting flies (S1 Table and S2 Table). The price of trypanocides was estimated at US$ 1.67 for >150 kg adult dose and delivery cost at US$ 5.56, bringing the cost per dose at US$ 0.036 / kg and thus US$ 28.9 per cattle / annum if administered quarterly. The cost could increase to US$ 115.6 / cattle / annum if trypanocides are administered monthly. Hence, if targeted areas are well-protected (traps, ITC, ITT), the trypanocide treatment would only be quarterly”.

In the discussion, the work of Shaw, 2009 has been referenced and compared with the findings on the cost analysis.

Reviewer #2: I found it difficult to engage with the study, and I believe this is down in part to the structure of the paper and some missing information.

I will give a small number of examples.

In the Materials and Methods, the section on the Nigeria case study (starting on page 8) contains a lot of information which discusses previous work, without being clear what the study you are reporting on does in this area (either materials or methods). What is the tsetse software? and tsetse plan? How do they work? Are they the implementations of your model? Or are values derived from the model being fed into them to generate outputs? Does using this software validate the model in some way? Or extend it's outputs (eg into the economics or operational planning).

Response: We have greatly improved the materials and methods, results and discussion sections. The prevalence of AAT in vertebrate host and vectoral capacity of biological and mechanical vector flies were important factors in elimination approach. Meanwhile, the southwest Nigeria field model validates our theoretical model because the all the biological and mechanical vectors were observed to harbour trypanosomes, in which our model showed that they have the potential to transmit the pathogen. The values from southwest Nigeria were entered in out Tsetse Plan software which automatically generates the scenario. Some of these field data were also important in our mathematical model. The whole concept was to validate our model with the field data to develop the best control approach.

I have spent a lot of time going backwards and forwards trying to find bits of information, or where things were first described. Possibly my own fault for printing the paper rather than viewing on screen. However, “To obtain R0 for the the model (2.1)-(2.11), the next generation matrix technique earlier described were utilised” appear on page 12 but the only previous reference to next generation matrix technique is in the abstract. So if it is described it isn't clearly labelled as such.

Response: The derivation of the next generation matrix has been improved upon and clearly stated. Many thanks

The model equations are referenced in numerous places as (2.1)-(2.11), but in my copy of the manuscript the equations appear to be labelled (0.1)-(0.11).

Response: Many thanks for the observation. We have corrected the labelling now.

There are a large number of grammatical and spelling mistakes, which I am unwilling to spend time on now given that the paper requires a substantial restructuring to separate material into its appropriate section and clarify how the different aspects of the study tie together. On next review, I would spend more time on the language.

Response: Many thanks for offering to help with the language. We will appreciate that. We also gave it to native English speakers for review to minimise the spelling mistakes.

---

## [Decision Letter · Decision Letter 1]

22 Sep 2020

PONE-D-20-18752R1

Mathematical modelling and control of African animal trypanosomosis with interacting populations in West Africa- could biting flies be important in maintaining the disease endemicity?

PLOS ONE

Dear Dr. Odeniran

Thank you for submitting your manuscript to PLOS ONE. After careful consideration, we feel that it has merit but does not fully meet PLOS ONE’s publication criteria as it currently stands. Therefore, we invite you to submit a revised version of the manuscript that addresses the points raised during the review process.

Many thanks for submitting your manuscript to PLOS One

It was reviewed by same two experts in the field who reviewed your original submission, and they have suggested some modifications be made prior to acceptance.

If you could write a response to reviewers, that will expedite review upon resubmission

I wish you the best of luck with your revisions

Hope you are keeping safe and well in these difficult times

Thanks

Simon

We look forward to receiving your revised manuscript.

Kind regards,

Simon Clegg, PhD

Academic Editor

PLOS ONE

Reviewers' comments:

Reviewer's Responses to Questions

**Comments to the Author**

1. If the authors have adequately addressed your comments raised in a previous round of review and you feel that this manuscript is now acceptable for publication, you may indicate that here to bypass the “Comments to the Author” section, enter your conflict of interest statement in the “Confidential to Editor” section, and submit your "Accept" recommendation.

Reviewer #1: All comments have been addressed

Reviewer #2: (No Response)

2. Is the manuscript technically sound, and do the data support the conclusions?

Reviewer #1: Yes

Reviewer #2: Partly

3. Has the statistical analysis been performed appropriately and rigorously? 

Reviewer #1: Yes

Reviewer #2: I Don't Know

4. Have the authors made all data underlying the findings in their manuscript fully available?

Reviewer #1: Yes

Reviewer #2: Yes

5. Is the manuscript presented in an intelligible fashion and written in standard English?

Reviewer #1: Yes

Reviewer #2: No

6. Review Comments to the Author

Reviewer #1: -Please consider rewriting sentences in lines 42, 67-70, 74, 191, 283, 425, 474-476, 493, 486, 497 for clarity. These sentences dont read fine to me

-fly insecticide resistance referred to in line 77 should be changed to biting flies insecticide resistance.... Insecticide resistance is not an issue for tsetse flies

-provide a table legend for table 1 explaining methods for parameter estimation in all instances where parameters were estimated

-Lines 199-205: please provide appropriate references to support the assumptions therein

-Line 219: ...moleculary...What do the authors mean here? which molecular techniques are these?

-Lines 220221-----highest prevalences of T.congolense and T.vivax...Provide these prevalence estimates in brackets after each trypanosome category

-Line 236...((FAA)... change to .....(FAA)

-Lines 405-406: Please provide a denominator for each of these costs. Overall elimination costs should be stated with the size of the operation area and the number of years it would take to achieve elimination

-Lines 463-464; provide an explanation why this is the case

-Lines 480-481: please give an explanation of how different livestock numbers, manpower etc are between Eastern uganda and West Nigeria

-Make sure that the colours used in the tsetse plan figures are the exact colours provided in the figure keys. Figures 2 and 3 seem to have discrepancies in the colours used and those provided in the figure keys.

Reviewer #2: The manuscript is a big improvement on the previous version. However, it still requires considerable work to improve clarity and make it publishable.

The actual structure of the study is unclear, and how the three components (the modelling, field work and the Tsetse Plan runs) tie together, how each informs the other. There are no citations for the Tsetse Plan software.

There is no consideration of variability in the parameters of the model (or used in other steps) - no sensitivity analysis or sampling of parameters. I acknowledge that this paper is the initial development of the model and maybe this will come in later publications, but it should be acknowledged and discussed rather than presenting the outcomes as definitive answers.

Is there evidence that tsetse flies are not required to maintain trypanosome populations? Are there no stages of the lifecycle that require this biological vector? Otherwise, removal of tsetse would result in removal of disease even if biting flies were present (although maybe with some lag).

I have made quite a few suggestions for minor changes in the attachment, however a more thorough re-write would still be appropriate.

7. PLOS authors have the option to publish the peer review history of their article (what does this mean?). If published, this will include your full peer review and any attached files.

Reviewer #2: No

---

## [Author Response · Author response to Decision Letter 1]

18 Oct 2020

Mathematical modelling and control of African animal trypanosomosis with interacting populations in West Africa- could biting flies be important in maintaining the disease endemicity?

Abstract

p.2, l.24 and l.46 trypanosomosis and trypansomiasis used, I’d choose one and stick to it. If you there is a reason for using trypanosomiasis in the keywords, why not use it everywhere?

Response- We have used trypanosomosis all through the manuscript.

p.2, l.26 assuming that AAT is regarded as a single disease despite the multiple causative agents, change ‘AAT...are a major threat...’ to ‘AAT...is a major threat...’

Response- Correction has been made as suggested by the reviewer.

p.2, l.33 change ‘...solving the system of the ordinary...’ to ‘...solving the system of ordinary...’

Response- We have corrected the sentence.

p.2, l.34 I don’t understand how R0 < 1 means you have to control the vectors, doesn’t R0 < 1 mean the disease is dying out? I think this part of the abstract is maybe just confused, or lacking some context?

Response- We have corrected the sentence as “The R0 < 1 in the formulated model indicates the elimination of AAT.”

p.2, l.42 change ‘individual or together’ to ‘individually or together’

Response- It has been corrected to individually or together

Introduction

p.3, l.52 ‘There are other trypanosome species which are also of significant importance to the livestock herd such as T. evansi and T. simiae.’ Are these species being ignored in the modelling? Or are all trypanosomes being handled together? in which case I’m not clear why these are differentiated in the text from the ‘major’ three. Having seen p.4, l.99, I would drop this sentence entirely, being so prominent at the start of the introduction it merely raises questions about something you aren’t interested in here.

Response- The parasites considered in the model were T. congolense, T. vivax and T. brucei brucei has observed by the reviewer.

p.3, l.58 delete ‘Generally,’ and delete ‘annual’

Response- It has been corrected.

p.3, l.60 I think Tabanidae and Muscidae are families, not genera.

Response- Many thanks. It has been changed to families

p.3, l.61 ‘AAT thresholds in various homesteads’, I’m not sure about the use of ‘thresholds’ here, maybe ‘levels’ would be better.

Response- Thresholds has been replaced with levels

p.3, l.62 change ‘...free of tsetse files, where biting flies...’ to ‘...free of tsetse files but where biting flies...’

Response- The word “where” has been included in the sentence.

p.3, l.67 change ‘Changes in vector distribution map...’ to ‘Changes to the vector distribution map...’

Response- It has been corrected to “Changes to the vector distribution map”

p.4, l.78 change ‘...areas are struggling...’ to ‘...areas is struggling...’

Response- It has been corrected to “ . . . areas is struggling”. . .

p.4, l.81-83 Do they consider biting flies at all? If not, I’d suggest something more like ‘Existing predictive models of African animal trypanosomiasis only consider the biological vectors Glossina spp., ignoring the possible importance of mechanical vectors [19,20,21,22]’.

Response- The sentence has been corrected to “Existing predictive models of African animal trypanosomiasis only consider the biological vectors Glossina spp., ignoring the possible importance of mechanical vectors [19,20,21,22]”.

p.4, l.85 some examples of programmes ignoring biting flies and, if there are any, programmes accounting for them would strengthen this. I don’t know, but I would guess that no programmes actively seek to eliminate biting flies, but the use of (eg) sprayed insecticides may impact on them as well as the tsetse.

Response- Programmes like sterile insect techniques, aerial spraying of insecticides along tsetse pockets (e.g. north-eastern Nigeria in 1967, 1500 km area in Lafia, Nasarawa State in Nigeria in the 1980s), excluded biting flies and only targeted tsetse flies. These examples (SIT and selective aerial spraying) have been included in the text.

p.4, l.87 And the significance of the variation in daily routines for control is what?

Response- The significance of variation in daily routines for control is the indirect impact on resistance to insecticides by these set of flies.

p.4, l.91 the models left knowledge gaps? Or the models suffered as a result of knowledge gaps? I suspect the latter, but I think the sentence implies the former.

Response- The sentence has been modified to mean that AAT models had suffered as a result of knowledge gaps.

p.4, l.99+ I’m not sure about this last paragraph. Partly because I’m not clear that this is what has been done in the study, I don’t see how you’ve evaluated the possibility of the model (I presume you would mean the probability of the model?).

Response- The sentence has been modified to "We also evaluated the probability of the model in the preliminary report conducted in southwest Nigeria and explained the reality of its elimination".

Materials and Methods

p.5, l.112 This is the first reference to bovine trypanosomosis. Why not stick to AAT throughout the manuscript (but making it clear in the intro that your primary interest is in the impact on the the cattle industry)? Do the wildlife and ruminant hosts not get the disease too?

Response- In line 11-112, we have modified the sentence to read "The model targets cattle, being mostly affected and having serious impact on food security in Africa". Each subsection introduces the host(s) considered in the model.

p.5, l.115-6 change ‘...the cattle input rate of new individuals entering the population. μk and τk are the...’ to ‘...the rate of new individuals entering the population, while μk and τk are the...’

Response- The sentence has been corrected to “the rate of new individuals entering the population, while μk and τk are the...” as suggested by the reviewer.

p.5, l.117 there are a lot of new terms in here, you should be referencing the table in which all these parameters are defined.

Response- We have referenced Table 1 for the parameters

p.6, l.126 do you mean ‘domestic livestock cycle’ rather than ‘domestic cycle’?

Response- Many thanks. We have corrected it to domestic livestock cycle.

p.6, l.126 sentence beginning ‘This peculiar...’, is there evidence to support this? If so reference it, but maybe in the introduction rather then M&M, alternatively if this is an hypothesis I don’t think it should be here. Does it mean that the choice of breeds for use in West Africa results in this situation?

Response- African dwarf breeds of goats and sheep are trypanotolerant compared to some other breeds. Most owners that raise them at subsistence level do not know of trypanosomosis because infection is majorly at subclinical level. However, they do come down to infection when the parasitaemia is high. Hence, we have referenced Isaac et al., 2017. For further reading, Behnke et al., 2011.

Are the cattle and ruminant breeds used elsewhere in Africa very different? Do they not support trypanosomes? This sentence seems out of place here.

Response- Breed selection is an important control strategy for AAT. There are some breeds (Zebu breeds of cattle) that are more susceptible than others (Taurine breeds). All are raised in Africa, over time breed selection is important in areas of heavy tsetse challenge. 

p.6, l.132 and l.141 again there are new undefined parameters, refer to the appropriate table of definitions.

Response- The parameters have now been referenced as table 1

p.7, l.157 change ‘climatic dependent’ to ‘climate-dependent’

Response- It has been corrected.

p.12, l.200 change ‘...treated cattle...’ to ‘...treating cattle...’

Response- It has been corrected.

p.12, l.199 sentence beginning ‘Already infected...’ I don’t fully understand this sentence, but I’m sure there is something wrong with it, may just be the structure of it makes it unclear.

Response- The sentence has now been modified, beginning with “Infected cattle are moved to the recovered class. . .”

p.12, l.202 what is a trypanocides safety period?

Response- It has been corrected to trypanocides withdrawal period.

p.12, l.211 The tsetse software is Tsetse Plan? If so, use this name and stick to it. Is this the Tsetse

Plan software available from Tsetse.org? Please provide a reference to the software being used, unless it is bespoke, in-house software in which case you should write more about it – here or in the Supplementary material.

Response- Yes, it is Tsetse Plan available from tsetse.org. We have given details on the software.

p.12, l.211 this heading, with a bracketed alternative heading, reads like you couldn’t decide what to call this section. And I’m struggling to think of a suitable heading that covers both parts (a) experimental and (b) Tsetse Plan evaluation. There is a lot in this section that looks like results and discussion of an experiment, but this is still the M&M section. Here you shoud be saying what you are doing (and maybe just mentioning any result that eg provides a parameter for your model), then have a corresponding section in the Results to present the outputs. I’m not clear that what is done in this section validates your model as stated.

Response- This section has been renamed “Assessment of control strategy in southwest Nigeria using Tsetse Plan”. A result section has been created, and all data values and results have been moved appropriately. This section explains how the field and experimental work were done, the generated results were used to create a field scenario in the Tsetse Plan to inform the best control strategies for elimination, including the cost. However, in the mathematical model, apart from using some of the field values, we analysed the probability and correctness by incorporating all the interacting populations involved (vertebrate host- cattle, small ruminants and wildlife, tsetse flies, biting flies- Stomoxys and tabanids and the parasites- T. vivax, T. congolense and T. b. brucei) using different mathematical tools. Our numerical analyses provide the best control approach (combining the field report, experimental results and Tsetse Plan reports) and suggesting likely outcomes of AAT situation if some measures are instituted 

p.13, l.219 I would change ‘screened molecularly’ to ‘screened for trypanosome DNA’ as (a) there could be other types of molecular screens and (b) the reader should not have to look up the reference to get that basic information.

Response- It has been corrected to ‘screened for trypanosome DNA’

p.13, l.237 Sentence ‘Therefore, an elimination approach...to control AAT inputs into Tsetse Plan’. Isn’t the goal of an elimination approach elimination, rather than provision of inputs to Tsetse Plan? Rewrite.

Response- It has been corrected to "Therefore, an elimination approach needs to consider approved insecticides (ITT) in both operational and adjacent areas in an integrated strategy to control AAT". We have moved this section to the results.

p.13, l.243 ‘The population density of the tsetse species was assumed to be medium from the overall report’ I assume medium is a setting in the tsetse plan software, but what does it mean as a density? 10 flies per km2? 100000?

Response- From the field work, we observed that the number of tsetse per square kilometre is estimated to be around 300 - 1000. This means that the density class in the input stage of the Tsetse Plan is categorised as medium. It also indicates that using Nzi traps without odour with average daily catches of both sexes ranging between 3-10 flies.

p.13, l.240 I think this paragraph is listing settings used for the Tsetse Plan software. How about a table accompanied by a more succinct statement highlighting those of most interest?

Response- We have improved this section as suggested by the reviewer.

p.13, l251 sentence beginning ‘Wildlife is...’ does this indicate very low wildlife reservoirs? Only if the grazing areas and wildlife-free areas are more-or-less the same ones. If all the 70km2 are grazing that means there could be a considerable wildlife reservoir in 30km2, couldn’t there? Or am I misreading what is there?

Response- The 30 km2 area that do only contain approximately five wildlife population. This means that the density is low. Besides, ruminants do not graze in most of these areas.

p.14-15 I’m struggling to see how the use of Tsetse Plan or the experiment validates your model. The experimental portion provides support for inclusion of biting flies, and provides parameter estimates as inputs to your model, but beyond that I’m confused. And also about how all of this is materials and methods. I’d like to see some kind of schema that explains how your model, the experiment and the Tsetse Plan run(s) link together (what data, parameters etc go from one part to another).

Response- We have included a brief statement on the linkages between the field/experimental, Tsetse Plan and the mathematical model provide the results and conclusion of this work. While Tsetse Plan gives the results on implementation and cost control directly from the field scenario in southwest Nigeria, mathematical model provides a more holistic result on elimination of the disease in West Africa.

Results

p.17, l.303 doesn’t the probability of a fly surviving a feed involve the probability it feeds on an untreated host too? And that sentence need rewriting, eg what is ‘feeds off and on treated hosts’?

Response- We have rephrased the sentence, it is now written as "The probability of vector fly surviving a feed is therefore, the product of the probabilities it feeds on untreated hosts and feeds on treated hosts and survives that meal".

p.17, l.304 did you predict or assume that flies feed off all cattle at random?

Response- The word predicted as been changed to assumed.

p.17, l.305 change ‘...systems widely...’ to ‘...systems being widely...’

Response- It has been corrected.

p.20, (no line given) rewrite the two lines beginning ‘In areas where...’, I don’t get what the bracket is about, there is is a subscript that isn’t fully subscripted and the biting flies are abundant (I think).

Response- The section has been corrected and written as “In areas where tsetse flies were absent (T. vivax showed R02 > 1 in the cattle population, when infected wildlife hosts are present), and biting flies (tabanids and stomoxyines) are abundant, the basic reproduction number of AAT increases”.

p.21, l.319 Does the mathematical model confirm the field-data? Didn’t the field-data inform the construction of the mathematical model? This seems a bit circular to me.

Response- We have corrected this sentence. It is now written as “The field-data from southwest Nigeria entered for the mathematical model generated valuable results applicable to West African countries”. 

p.21, l.324? Remove inserted.

Response- It has been removed.

p.21, l.325? Remove significantly.

Response- It has been removed.

p.21, l.?? change ‘...poses a risk on...’ to ‘...poses a risk to...’. I don’t think I understand the whole of that sentence ‘The possibility of continuous treatment...’, it is unclear to me what the risk to the herd is from treatment in the presence of infected vectors, please clarify.

Response- The risk here referred to trypanocide resistance risk. We have modified the sentence as “The possibility of continuous trypanocide treatment in the presence of infected vector flies poses trypanocide resistance risk to the cattle herd”. Many thanks

p.22, l.327 I assume that there is no requirement in the trypanosome lifecycle to pass through a biological, rather than just mechanical, vector? The lifecycle stages that take place in tsetse can occur just as well elsewhere? Any evidence to support this?

Response- Yes, if the mechanical vectors are present, some species of trypanosomes such as T. vivax can be transmitted without tsetse flies. We have included examples, “For instance, T. vivax has managed to maintain itself in South America where tsetse flies are absent, with Tabanidae and Stomoxys acting as mechanical vectors [Reis et al., 2019]. Similarly, Anene et al. [7] observed that T. vivax was maintained in the flock by tabanids in tsetse-free areas in Nigeria”. 

p.23, l.334 change ‘It is suffice...’ to ‘It is sufficient...’

Response- It has been corrected.

p.23, l.336 is (0.20) an acceptable equation reference in PLOS One? It looks ugly to me, shouldn’t it at least be preceded by eq. or equation?

Response- We have included the word “equation”. Now written as (equation 0.20).

p.24, l.360 change ‘...that susceptible...’ to ‘...that the susceptible...’

Response- It has been corrected.

p.24 The two paragraphs that begin ‘The graph-based behaviour...’, could they be rewritten in such a way as to make differentiate them from one another a bit more. They are very repetitive.

Response- We have modified the second “graph-based behaviour” line to avoid repetition. It is now written as “The illustrated graph of the tsetse fly populations showed that the magnitude of susceptible tsetse fly decreases as a result of infection from infected cattle and the use of insecticide (Fig 3C)”.

Fig. 3 I can’t see a legend so it is unclear what lines refer to what.

Response- The legend has been inserted.

p.26, l.397 change ‘We observed...’ to be ‘We estimated...’

Response- It has been corrected.

p.26, l.399 change ‘Meanwhile, if...’ to ‘However, if…’

Response- It has been corrected.

p.26, l.400 change ‘...for longer period, cost...’ to ‘...for a longer period, the cost...’

Response- It has been corrected.

p.26, l.401 this is the first (and only?) reference to ‘fly belt zones’ I would recommend avoiding introducing additional terminology at this late point in the paper. You must have described the areas inhabited by flies earlier, reuse that nomenclature.

Response- We have removed the word “fly belt zone” at this point as suggested by the reviewer. Many thanks.

p.26 The final three sentences on this page, do the all relate to measures to improve the success rate score? It reads as if the first does and then the other two are things that were also done, they should form more of a list. And what does the success rate score of 70% in Tsetse Plan mean?

Response- We have improved these sentences to avoid ambiguity. It now reads, “To improve the success rate, the insecticidal approach (ITT) must be continuously maintained both within and outside the operational area. Barriers between the domestic and sylvatic areas need to be made active, while periodical assessment of cattle blood should be a routine practice. In the presence of vector flies, quarterly use of trypanocides and ITC on ruminants in a structured manner across the study areas need to be instituted”.

Discussion

p.27, l.424 spelling ‘conditions’

Response- It has been corrected.

p.27, l.425 spelling ‘Tsetse’

Response- It has been corrected.

p.27, l.427 sentence beginning ‘In fact, Rogers...’ needs rewriting. I haven’t read the reference, but is it the model or the paper that expresses a limitation? And by this do they mean difficulty of including biting flies in the model? The sentence is unclear. I think, and I’m guessing at the meaning a bit, that it something like ‘The model of Rogers [19], which was the prototype of subsequent models, omitted biting flies because of the difficulties of including them’ or maybe ‘including them reliably’ if the problem is one of parameterising the models.

Response- We have improved the sentence to enhance the understanding of our readers. It now reads, “In fact, Rogers model expresses limitation of considering biting flies in a field-based model because its importance relative to cyclical transmission has not been fully established in the field as at then, which thereafter remained prototype for subsequent models [19,27]”. This was established in the discussion section (third paragraph, page 210) of Rogers’ model.

p.27, l. 430 change ‘helps’ to ‘help’.

Response- It has been corrected.

p.27, l.440 sentence (paragraph?) needs rewriting, it implies that complications exist as a result of the model, whereas they are practical considerations which are in place. Either the model should consider them, or perhaps the outcomes of modelling could influence the strategy in the future.

Response- We have rephrased the sentence. It now reads, “However, the model considers some areas in West Africa which had few or no tsetse flies present in which trypanocides are only given when cattle show signs of disease”.

p.28, l.450 Doesn’t this also impact tsetse? Or does their strange, for an insect, reproduction make this less likely? I guess so.

Response- We have included a sentence which reads, “This resistance is less likely in tsetse flies because they are K-strategists, in which case they have low reproducing capacity with very high success rate”. 

p.28, l.464 I think you mean a low rate of infection could persist rather than a slight infection. Or a small population of tsetse through reintroduction/migration result ing on-going infections.

Response- We have modified the sentence to read that a low rate of infection could still persist. 

p.28, l.468 ‘...except cattle are...’ implies to me that all cattle are kept in intensive, optimal conditions. Is this right? Or do you mean ‘...except where cattle are...’?

Response- Many thanks. We have corrected the sentence to except where cattle are.

p.28, l.473 change ‘However, it is expected to use artificial baits and targets (ITT) in the other parts where cattle do not visit’ to ‘However, the use of artificial baits and targets (ITT) is expected in the parts where cattle do not visit’

Response- Response- It has been corrected as suggested by the reviewer.

p.29, l.479 change ‘...costs...was...’ to ‘...costs...were...’

Response- It has been corrected.

p.29, l.483 I’m not sure what this sentence means. The tsetse is infected, bites a treated host and transmits to the host before dying. So what is the bit about the tsetse becoming infected through feeding about? And then I don’t see how the following sentences carry on from this.

Response- The sentence has been modified properly. It now reads, “It is expected that infected tsetse population would reduce due to the insecticide treatment, rather than the tsetse becoming infected through feeding”.

p.30, l.517 I think ‘sufficiently eliminated’ should be ‘eliminated’ or ‘sufficiently controlled’.

Response- Response- It has been corrected to sufficiently controlled.

Sensitivity analysis? Variability in parameters?

Response- We did not conduct sensitivity analysis, which could be done with consideration of other variables in future studies. We conducted parameter estimation which has been included in the data analysis. It is written as, “The estimation of parameters was achieved using the least squares method in Excel solver [34], with a view to minimising summation of squared errors given by ∑(Y(t; p)-Xreal)2 subject to the AAT model (0.1)-(0.16) where Xreal is the field reported data, and Y(t; p) represents the solution of the model corresponding to the number of active cases divided by time t with the set of estimated parameters denoted by p”.

Note- Tsetse team members have been acknowledged.

REVIEWER’S COMMENT

Reviewer #1: 

-Please consider rewriting sentences in lines 42, 67-70, 74, 191, 283, 425, 474-476, 493, 486, 497 for clarity. These sentences dont read fine to me

Response- The whole manuscript has been edited again by the English-speaking co-authors on the manuscript. 

-fly insecticide resistance referred to in line 77 should be changed to biting flies insecticide resistance.... Insecticide resistance is not an issue for tsetse flies

Response- This has been corrected

-provide a table legend for table 1 explaining methods for parameter estimation in all instances where parameters were estimated

Response- A comment on parameter/data estimation has been provided under data analysis section. We observed that it fits well in that section.

-Lines 199-205: please provide appropriate references to support the assumptions therein

Response- This has been provided

-Line 219: ...moleculary...What do the authors mean here? which molecular techniques are these?

Response- It has been written as “screened for trypanosome DNA”

-Lines 220221-----highest prevalences of T.congolense and T.vivax...Provide these prevalence estimates in brackets after each trypanosome category

Response- The study reported highest prevalence of T. congolense 11.0% (95%CI: 8.5–14.2) and T. vivax 14.7% (95%CI: 10.96–19.49) in the wet and dry seasons, respectively.

-Line 236...((FAA)... change to .....(FAA)

Response- This has been corrected.

-Lines 405-406: Please provide a denominator for each of these costs. Overall elimination costs should be stated with the size of the operation area and the number of years it would take to achieve elimination.

Response- A sentence that reads below has been included, “The overall elimination costs was US$ 17,963,490 per annum in an area of 78,000 km2 of southwest Nigeria, in which elimination could be achieved within three years of consistent control measures”.

-Lines 463-464; provide an explanation why this is the case

Response- This has been thoroughly explained in the manuscript. Without, ITT the model shows that areas beyond the target areas could pose risk from infected transmitting vectors.

-Lines 480-481: please give an explanation of how different livestock numbers, manpower etc are between Eastern uganda and West Nigeria

Response- We have earlier stated that “This could be attributed to the number of livestock, management practices, manpower and density of flies”. The infected fly density and management practices in these two areas differ and could be responsible for differences in the cost estimated.

-Make sure that the colours used in the tsetse plan figures are the exact colours provided in the figure keys. Figures 2 and 3 seem to have discrepancies in the colours used and those provided in the figure keys.

Responses- For the tsetse plan figures, we used generated figures directly from the software. Legend has been included for figure 3. The colours were generated from the software used.

Reviewer #2: 

The manuscript is a big improvement on the previous version. However, it still requires considerable work to improve clarity and make it publishable.

Response- We have greatly improved the manuscript now.

The actual structure of the study is unclear, and how the three components (the modelling, field work and the Tsetse Plan runs) tie together, how each informs the other. There are no citations for the Tsetse Plan software.

Response- We have included a brief statement on the linkages between the field/experimental, Tsetse Plan and the mathematical model provide the results and conclusion of this work. While Tsetse Plan gives the results on implementation and cost control directly from the field scenario in southwest Nigeria, mathematical model provides a more holistic result on elimination of the disease in western Africa. This section explains how the field and experimental work were done, the generated results were used to create a field scenario in the Tsetse Plan to inform the best control strategies for elimination, including the cost. However, in the mathematical model, apart from using some of the field values, we analysed the probability and correctness by incorporating all the interacting populations involved (vertebrate host- cattle, small ruminants and wildlife, tsetse flies, biting flies- Stomoxys and tabanids and the parasites- T. vivax, T. congolense and T. b. brucei) using different mathematical tools. Our numerical analyses provide the best control approach (combining the field report, experimental results and Tsetse Plan reports) and suggesting likely outcomes of AAT situation if some measures are instituted 

There is no consideration of variability in the parameters of the model (or used in other steps) - no sensitivity analysis or sampling of parameters. I acknowledge that this paper is the initial development of the model and maybe this will come in later publications, but it should be acknowledged and discussed rather than presenting the outcomes as definitive answers.

Response- All the analyses done were mainly mathematical. On the matter of sensitivity analysis, we would leave it for future scientific work. Many thanks

Is there evidence that tsetse flies are not required to maintain trypanosome populations? Are there no stages of the lifecycle that require this biological vector? Otherwise, removal of tsetse would result in removal of disease even if biting flies were present (although maybe with some lag).

Response-Some species of trypanosomes such as T. vivax can be transmitted without tsetse flies. We have included examples, “For instance, T. vivax has managed to maintain itself in South America where tsetse flies are absent, with Tabanidae and Stomoxys acting as mechanical vectors [Reis et al., 2019]. Similarly, Anene et al. [7] observed that T. vivax was maintained in the flock by tabanids in tsetse-free areas in Nigeria”. Not all species of trypanosomes require the tsetse fly phase to cause infection. Many thanks 

I have made quite a few suggestions for minor changes in the attachment, however a more thorough re-write would still be appropriate.

Response- All the suggestions have been adhered to. We have made the document a lot better.

---

## [Editor Report · Decision Letter 2]

3 Nov 2020

Mathematical modelling and control of African animal trypanosomosis with interacting populations in West Africa- could biting flies be important in maintaining the disease endemicity?

PONE-D-20-18752R2

Dear Dr. Odeniran,

We’re pleased to inform you that your manuscript has been judged scientifically suitable for publication and will be formally accepted for publication once it meets all outstanding technical requirements.

Kind regards,

Simon Clegg, PhD

Academic Editor

PLOS ONE

Additional Editor Comments:

Many thanks for resubmitting your manuscript to PLOS One

As you have addressed all the comments, and the manuscript reads well, I have recommended it for publication

You should hear from the Editorial Office soon

It was a pleasure working with you and I wish you all the best for your future research

Hope you are keeping safe and well in these difficult times

Thanks

Simon

---

## [Editor Report · Acceptance letter]

10 Nov 2020

PONE-D-20-18752R2 

Mathematical Modelling and Control of African Animal Trypanosomosis with Interacting Populations in West Africa- Could Biting Flies be Important in Main taining the Disease Endemicity? 

Dear Dr. Odeniran:

I'm pleased to inform you that your manuscript has been deemed suitable for publication in PLOS ONE. Congratulations! Your manuscript is now with our production department. 

Kind regards, 

on behalf of

Dr. Simon Clegg 

Academic Editor

PLOS ONE